# Atmospheric measurements of the terrestrial $O_2$:$CO_2$ exchange ratio of a mid-latitude forest

Mark O. Battle[1], J. William Munger[2], Margaret Conley[1], Eric Sofen[1], Rebecca Perry[1], Ryan Hart[1], Zane Davis[1], Jacob Scheckman[1], Jayme Woogerd[1], Karina Graeter[1], Samuel Seekins[1], Sasha David[1], and John Carpenter[1]

[1]Dept. of Physics & Astronomy, Bowdoin College, Brunswick ME 04011-8488 USA
[2]School of Engineering and Applied Sciences, Harvard University, Cambridge, MA 02138 USA

**Correspondence:** Mark O. Battle (mbattle@bowdoin.edu)

**Abstract.**

Measurements of atmospheric $O_2$ have been used to quantify large-scale fluxes of carbon between the oceans, atmosphere and land since 1992 (Keeling and Shertz, 1992). With time, datasets have grown and estimates of fluxes have become more precise, but a key uncertainty in these calculations is the exchange ratio of $O_2$ and $CO_2$ associated with the net land carbon sink ($\alpha_B$). We present measurements of atmospheric $O_2$ and $CO_2$ collected over a six-year period from a mixed deciduous forest in central Massachusetts, USA (42.537°N, 72.171°W). Using a differential fuel-cell based instrument for $O_2$ and a non-dispersive infrared analyzer for $CO_2$, we analyzed airstreams collected within and $\sim 5$ m above the forest canopy. Averaged over the entire period of record, we find these two species covary with a slope of $-1.081 \pm 0.007$ moles of $O_2$ per mole of $CO_2$ (the mean and standard error of 6-hour periods). If we limit the data to values collected on summer days within the canopy, the slope is $-1.03 \pm 0.01$. These are the conditions in which biotic influences are most likely to dominate. This result is significantly different from value of -1.1 widely used in $O_2$-based calculations of the global carbon budget, suggesting the need for a deeper understanding of the exchange ratios of the various fluxes and pools comprising the net sink.

tmospheric $O_2/N_2$, oxidative ratio, terrestrial carbon sink

## 1 Introduction

Since the pioneering work of Keeling and Shertz (1992), measurements of the abundance of atmospheric $O_2$ and $CO_2$ have been used extensively for constraining fluxes of carbon to and from the land biosphere and the oceans. In a conceptual form, the budgets for atmospheric $O_2$ and $CO_2$ can be written

$$\frac{dO_2}{dt} = \alpha_{ff} \, f_{ff} + \alpha_B \, f_{land} + Z_{ocean} \tag{1}$$

$$\frac{dCO_2}{dt} = f_{ff} + f_{land} + f_{ocean} \tag{2}$$

where $f_{xx}$ is the flux of $CO_2$ from reservoir $xx$ to the atmosphere, $\alpha_{ff}$ is the global average oxidative ratio of fossil fuels and $\alpha_B$ is the average oxidative ratio of the global net land carbon sink. $Z_{ocean}$ describes the outgassing of $O_2$ from the ocean due to warming. Knowing $f_{ff}$ and $\alpha_{ff}$ from industrial inventories, calculating $Z_{ocean}$ from changes in ocean heat storage, and measuring the changes in $O_2$ and $CO_2$, we can solve these two equations for $f_{land}$ and $f_{ocean}$ in terms of $\alpha_B$ (for a rigorous treatment of these equations, see for example Keeling and Manning (2014)). This $O_2$-based method has become increasingly sophisticated with refinements in methodology, and increasingly precise with better instruments and ever-longer datasets (Keeling and Manning, 2014). Nonetheless, all estimates of global carbon fluxes based on atmospheric $O_2$ require a value for $\alpha_B$.

The molar ratio of $O_2$ released to the atmosphere and $CO_2$ removed from it during photosynthesis (one contribution to $\alpha_B$) will depend on the type of organic material being synthesized. This ratio is set by the relative amounts of carbon and hydrogen in the material, as well as the amount and source of any nitrogen included. Similarly, the ratio of $CO_2$ release and $O_2$ uptake during respiration depends on the composition of the material being respired and whether nitrogen is oxidized to nitrate. For example, synthesis and respiration of the simplest sugar (glucose: $C_6H_{12}O_6$) will have a flux ratio of $1.0$. However, if plants are making shoots and other nitrogen-rich tissue, the oxidative ratio might range from $1.0$ to $1.26$, depending on the source of the nitrogen (nitrate vs. ammonium) (Bloom et al., 1989). Clearly, any value of $\alpha_B$ that is used in a large-scale calculation of carbon fluxes must represent the net imbalance of many processes in many ecosytems over long times.

One approach to this problem is to survey the elemental ratios of various biogenic materials. This was done by Severinghaus (1995); Randerson et al. (2006); Masiello et al. (2008); Hockaday et al. (2009); Worrall et al. (2013) and Gallagher et al. (2017). Based on a variety of techniques and a range of materials, estimates of $\alpha_B$ at present seem to be around $1.05\pm0.1$(Keeling and Manning, 2014). Propagating this uncertainty to the global land-ocean partition of carbon uptake corresponds yields a value of $\pm0.1\,\mathrm{PgCyr}^{-1}$.

One method that can illuminate the processes responsible for $\alpha_B$, and the one we have chosen, was first published by Seibt et al. (2004); Sturm et al. (2005) and Stephens et al. (2007). It uses direct measurements of $O_2$ and $CO_2$ changes in ambient air within, and close to, an ecosystem. Given sufficient precision and temporal resolution, the covariation of these two atmospheric components should reflect the exchange ratio for the particular mixture of fluxes to/from the ecosystem present during the study period. We refer to this quantity as the local biotic exchange ratio (LBER), recognizing that it is closely related to, yet distinct from, $\alpha_B$. $\alpha_B$ may turn out to be close to measured values of LBER, but it is a distinct quantity by definition: $\alpha_B$ is the ratio of global net $CO_2$ and $O_2$ exchanges from *all* terrestrial biological processes.

The most groundbreaking feature of the work of Stephens et al. (2007) was the use of a newly-developed, continuous $O_2$ analyzer deployed at the study site, allowing precise, high-frequency measurements. These techniques have the potential to better characterize contributions to $\alpha_B$ as well as improve our understanding of the underlying physiology of the ecosytem generating the fluxes.

Following Stephens et al. (2007), we have used a fuel-cell-based differential $O_2$ analyzer and a non-dispersive infrared (NDIR) $CO_2$ analyzer to make quasi-continuous measurements of the air in and above the canopy at the Harvard Forest Environmental Measurement Site (Urbanski et al., 2007). In following sections of this manuscript, we begin with a description of the instrument and our methods of data collection and analysis. We then present data acquired between 2006 and 2013 and

make a first effort at interpreting these data in the context of both the ecosystem's process-specific ratios and the response of the ecosystem to environmental controls.

A note on units: Because $O_2$ is a major atmospheric constituent, its abundance is customarily reported in "per meg" defined as

$$\delta \left( O_2/N_2 \right) = \left( \frac{O_2/N_2}{O_2/N_{2 \, st}} - 1 \right) \times 10^6 \qquad (3)$$

where $O_2/N_{2 \, st}$ is $O_2/N_2$ for the standard gas on which the scale is based. This avoids complications arising from dilution effects (Keeling et al., 1998). Because our measurements are made using an instrument that reports differences in apparent mole fraction, we convert to per meg using Eq. 3 of Stephens et al. (2007). Finally, when working with exchange ratios, we convert per meg to "ppm equivalent" by multiplying per meg by 0.209500 (the $O_2$ mole fraction of atmospheric air) (Keeling et al., 1998). When using ppm equivalent for $O_2$ and $\mu mol \, mol^{-1}$ (or ppm) for $CO_2$, the synthesis of glucose (described above) would increase $O_2$ and decrease $CO_2$ by an equal number of units, directly translating our measurements to the underlying stoichiometry, as desired.

## 2   Methods

In August 2005 we installed a measurement system at the Harvard Forest Environmental Measurement Site (EMS). Located at 42.5377°N, 72.1714°W and 340 m above sea level, we collected air from two intake tubes mounted on a tower 7.50 m and 29.0 m above the ground. The lower intake is within the forest canopy, the upper about 5 m above it. The tower and site are described in more detail by Urbanski et al. (2007). Our system is very similar to the one installed at WLEF in Wisconsin and documented very thoroughly by Stephens et al. (2007), so we focus primarily on differences from the WLEF instrument in our description here.

### 2.1   System design and operation

A schematic of the system is shown in Fig. 1. Unless otherwise mentioned, all wetted surfaces are stainless steel. At both points of collection, air is drawn in through downward-facing aspirated inlets (R.M. Young, Part #43408) to reduce thermal fractionation (Blaine et al., 2006). A nominal flow rate of $50 \, cm^3 min^{-1}$ is maintained at all times in both lines using pumps with viton diaphragms and PTFE valves (KNF Neuberger N05-ATI) in concert with mass flow controllers (MKS M100B). Tubing on the tower ($\frac{3}{8}'' O.D.$ Dekabon®) is routed into the instrument enclosure without local low spots to minimize buildup of condensed water. Within the building, water is removed from the airstream using a series of three coaxial stainless steel traps. The first is kept at 5° C with a Peltier cooler (M&C model ECP2000-SS), while the second and third traps are filled with borosilicate glass beads and immersed in a heat-transfer fluid (Syltherm-XLT®) held at $-90°$ C using a mechanical refrigeration system (FTS VT490). Filters with a nominal pore size of 7 $\mu m$ (Swagelok SS-6F-7) protect downstream equipment from particulates. Since our analyzers can only measure a single airstream, the high and low intake lines are alternately sent to

analysis or waste. The intake selector valve is a pneumatically actuated, crossover ball valve with PTFE packing (Swagelok SS-43YFS2), switched every 720 seconds.

Both the $O_2$ and $CO_2$ analyzers make differential measurements of the chosen stream of ambient air relative to a cylinder of dried air (the working tank, WT). Data are recorded at 1 Hz. Periodically, the ambient air stream is diverted to waste and the WT is analyzed against one of four calibration tanks: HS, LS, MF and LF. Details of the various tanks are given in Table 1. The frequency of the calibration runs was determined in part from pre-deployment tests in which we observed instrumental drift when running pairs of (sacrificial) tanks against one another. Other considerations include consumption of standard gases and lost time for atmospheric measurements.

The ambient air stream (high or low) or the calibration tank is chosen for analysis with a pair of electrically actuated 6-port selector valves (VICI EMT2SF6MWE and EMT2SDMWE). From these valves it passes through another mass flow controller, the NDIR $CO_2$ analyzer (Licor LI-7000), a $0.5~\mu m$ filter (Swagelok SS-6F-05), a needle-tipped metering valve (Swagelok SS-4BMG), a changeover valve assembly (four Numatics TM101V24C2 solenoid valves on a custom-built aluminum manifold), and the fuel-cell based $O_2$ analyzer (Sable Systems Oxzilla II with customized stainless steel internal plumbing), then exits through a mass-flow meter (Honeywell AWM3100V). Absolute pressures are recorded upstream of the $CO_2$ analyzer in three locations (Omega PX303-030A10V) and within the Oxzilla (see below).

The response of the Licor $CO_2$ analyzer depends on the pressure difference between the two cells. Thus, we use a differential manometer and electronically controlled metering valve with active feedback (MKS 223BD, 247D and 248) to ensure balanced pressures in the Licor.

The response of the Oxzilla $O_2$ analyzer depends on both pressure and flow differences between the cells. To provide sufficient control of these parameters, we have two metering valves upstream of the Oxzilla and one downstream. We use the metering valves (as well as the active pressure-difference control upstream of the Licor) to tune the system so that pressures and flows are as close to equal as possible regardless of the state of the changeover valve (see below). For this tuning, and subsequent monitoring of the system, we use flow data from the Honeywell meters and pressure data recorded within the Oxzilla immediately downstream of the cells. Prior to deployment, we modified the Oxzilla's internal plumbing and electronics so that pressure transducers (Honeywell SCX15N) were teed off of each of the outlets of the fuel cells. The two transducers are read out with the single high-precision digitizer built into the Oxzilla motherboard using a multiplexer to alternately select the transducer signal sent to the digitizer.

To further improve measurement precision of $O_2$, we follow Stephens et al. (2007) and use a changeover valve to alternate the fuel cells into which the two gas streams flow. This allows for a difference-of-differences calculation that eliminates cell-to-cell biases in the instrument. However, the elimination of bias occurs only if that bias varies little during the two measurement periods being considered. Achieving optimal precision would seem to require high frequency changeovers to reduce variation in the bias. However, each changeover is followed by a "dead time" when flows and instrument response need to settle. For a given dead time, more rapid changeovers mean fewer measurements are available for averaging, leading to a greater impact of random noise and thus, poorer precision.

We adopt a method for quantifying this trade-off and settling on an optimal changeover frequency that was developed by Keeling et al. (2004). We determined dead time for our instrument by analyzing tank air (relative to a working tank) with the changeover running very slowly. We then measured the time after each changeover required for $O_2$ values to settle to within $1\sigma$ of the post-changeover equilibrium value. This "dead time" was 14 seconds for our instrument. We then determined the optimal

changeover time by analyzing two tanks of air against each other with the changeover valve disabled. We processed these measurements as if the changeover valve were operating, calculating difference-of-difference values for various (hypothetical) changeover rates, imposing a 14 s dead time. This exercise indicated an optimal changeover rate of 24 s between switches.

## 2.2    Data reduction

### 2.2.1    Atmospheric data

In order to allow the instrument to achieve equilibrium after the selector valves have changed state (at the end of a calibration run), we discard the first 432 s of atmospheric data after such an actuation.

The remaining atmospheric data are naturally divided into 12-minute subsets by the switching of the intake selector between high and low intake lines. We discard the first 432 s of data after an intake switch to allow time for the instrument to reëquilibrate after the disruption in flows and pressures. The surviving data are processed slightly differently for $O_2$ and $CO_2$.

For $O_2$, we break the surviving 288 s of data into 24 s blocks defined by the cycling of the changeover valve. We average the 1 Hz data collected between 14 s and 24 s after the valve changes state. We then calculate difference-of-difference values for consecutive pairs of averages. This typically yields 6 values. These are in turn averaged, yielding a single $O_2$ value for the chosen intake height at a time corresponding to the middle of the interval of data used in the average.

For $CO_2$, we consider every 1 Hz $CO_2$ value collected at the same time as the 1 Hz $O_2$ values that contribute to the final

$O_2$ mixing ratio for each of the 12-minute intake intervals. After the cuts for deadtime following valve actuations (gas selector, intake selector and changeover), there are 120 s of live data, so our $CO_2$ mixing ratio is an average of 120 measurements.

### 2.2.2    Calibration data

Roughly four times daily, atmospheric sampling is interrupted for 6-minute runs of calibration tanks. To more effectively use the information from these runs, we adopt a protocol that differs from our treatment of atmospheric data.

To flush stale gas from the lines and purge the regulators, air from the calibration tank is vented to waste for 6 minutes prior to measurement. Once the selector valves direct the calibration air to the analyzers, we consider all 6 minutes of $CO_2$ measurements. We look for a transition in the $CO_2$ value as tank air reaches the analyzer, displacing the atmospheric air that was in the optical cell. We find the time at which 70% of the eventual change in $CO_2$ has been achieved (the transition time).

To determine $O_2$ in these calibration runs, we calculate difference-of-differences values for each pair of changeover blocks as with our atmospheric data. Then, based on the timing of the transition in the $CO_2$ record, we average the last 3, 4, or 5 difference-of-difference values of the calibration run. The transition time typically occurs $\sim 2.6$ minutes after moving the

selector valve. In this case, we average 3 difference-of-difference pairs, corresponding to the last 2.4 min of data. This translates to averaging the last $\sim 40\%$ of each cal run (or $\sim 70\%$ of the post-transition data).

To determine $CO_2$, we average all $CO_2$ values collected at the same times as the values that were used in the $O_2$ calculation.

Our standards are calibrated on the Scripps S2 $O_2$ scale (Keeling et al., 1998, 2007) and the NOAA/WMO $CO_2$ scale (Zhao and Tans, 2006), thanks to the generous assistance of these laboratories. Thus it is on these scales that we report our measurements.

## 2.3 System performance

While we have made great efforts to calibrate our measurements to internationally accepted $CO_2$ and $O_2$ scales using our suite of reference tanks (Table 1), we defer a detailed treatment of long-term measurement stability and accuracy for a later publication. Our focus in this manuscript is on the local biotic exchange ratio, as expressed in a series of short-term averages of $CO_2$ and $O_2$ covariation. Thus, our primary concern is instrumental precision and stability over periods of roughly 6 hours.

To determine this instrumental precision, we use runs of calibration tanks (see Sec. 2.2.2).

There are two possible approaches: We could use the observed scatter in the 3, 4 or 5 difference-of-difference values for $O_2$ that are calculated during a 6-min run of a calibration tank, and likewise for the 60, 80 or 100 $CO_2$ measurements retained during the same run. These are the values plotted in Fig. 2. We refer to this metric as "scatter on one tank" (SOOT).

Not surprisingly, the performance of the instrument varies over time. At times, we could link a loss of precision to problems with an individual analyzer, while on other occasions, we found water (liquid or solid) in the system to be the cause. The source of the major degradation in the precision of both analyzers that develops in late 2012 and persists until the end of our dataset remains under investigation.

An alternative approach, and the one that we adopt here, uses the observed scatter in the difference of paired standard tanks that were run immediately following one another. The calculation of the difference between the HS and LS tanks run in succession emulates the treatment of the atmospheric data by comparing one gas stream (HS in place of atmosphere) to another (LS in place of a "calibration curve") without the involvement of the working tank. The scatter in the HS-LS difference is a conservative estimate of uncertainty in atmospheric data because it includes any variability that might be introduced by operation of the gas-selection valve, a valve that remains fixed in position during atmospheric sampling.

We find the the scatter in the HS-LS differences for 12-day periods is around $\pm 2$ ppm equivalent for $O_2$ and $\pm 0.2$ $\mu$mol mol$^{-1}$ for $CO_2$ until the middle of 2008. From 2012 onward, the scatter was $\pm 9$ ppm equivalent and $\pm 0.6$ $\mu$mol mol$^{-1}$.

Each HS and LS $O_2$ value was typically the average from 3 difference-of-difference pairs, while the $CO_2$ values were usually an average of 60 measurements. In contrast, a 12-minute atmospheric measurement on either the high or low intake is an average of 6 $O_2$ difference-of-difference pairs and 120 $CO_2$ measurements (Sec. 2.2.1). Thus, the atmospheric measurements should have uncertainties given by

$$\sigma_{O_2 \; atm} = \frac{\sigma_{O_2 \; HS-LS}}{\sqrt{2}} \frac{\sqrt{3}}{\sqrt{6}} \tag{4}$$

$$\sigma_{CO_2\ atm} = \frac{\sigma_{CO_2\ HS-LS}}{\sqrt{2}} \frac{\sqrt{60}}{\sqrt{120}} \tag{5}$$

Based on these equations, we believe that the atmospheric measurements carry uncertainties of $\sigma_{O_2\ atm} = 1$ ppm equivalent and $\sigma_{CO_2\ atm} = 0.1\ \mu mol\ mol^{-1}$ prior to mid-2010. For measurements beginning in 2012, the uncertainies were $\sigma_{O_2\ atm} = 4.5$ ppm equivalent and $\sigma_{CO_2\ atm} = 0.3\ \mu mol\ mol^{-1}$.

Our choice to work backwards from the observed scatter in HS-LS pairs may not be the optimal technique for extracting uncertainties, but we are confident it yields $O_2$ and $CO_2$ errors that bear the correct *relative* size. Thus, we use the value listed above when calculating slopes in our Deming regressions (Sec. 3).

## 3 Observations

The measurements we present here commenced on September 27, 2006 and continued until January 9, 2013. The record is
10 approximately continuous until the middle of 2010, when a series of hardware failures, due in part to a lightning strike, caused an extended hiatus in data collection. Measurements resumed in May of 2012 and continued without interruption through the end of that year, when failure of the oxygen analyzer led to a suspension of data collection.

Measurements of $O_2$ and $CO_2$ for the entire period of record are shown in Fig. 3, while Fig. 4 gives an example of a single day's cycle. The values shown here were derived from the raw data using the calibration runs and the algorithm described in
Sec. 2.2. They represent our best estimates of the $O_2$ and $CO_2$ content of the atmosphere on the Scripps S2 and NOAA/WMO scales, respectively. In these figures we can see a very gradual rise in $CO_2$ and drop in $O_2$ (mainly due to fossil fuel combustion), a strong diel cycle with inverse variation in $CO_2$ and $O_2$, and a great deal of variability on a range of time scales. Results from the upper and lower intakes are quite similar, but not identical.

While these records contain a wealth of information, our particular focus is on the covariation of $O_2$ and $CO_2$. We expect
to see different results from the high and low intakes, particularly at night. The lower intake is more likely to show stronger influences of soil and canopy fluxes, particularly at night when the atmosphere is more stable and the surface layer is more likely to be decoupled from the planetary boundary layer. For this reason, we separate the data from the high and low intakes, create day and night subsets for each intake, plot $O_2$ vs. $CO_2$ for each subset, perform a linear regression, and take LBER from the slope of the fit.

The 6-hour subsets making up each plot are 9am - 3pm (day) and 10pm - 4am (night). These intervals are chosen to be times of maximal and minimal coupling (day and night, respectively) between the air within and above the canopy. We define the intervals based on the climatological diel cycle in the friction velocity $u^\star$ measured at the EMS tower (Munger and Wofsy, 2017). Subsets with fewer than three measurements are ignored. Because uncertainties in both species are comparable, we use a Deming regression (Deming, 2011). Furthermore, to reduce any influence of non-Gaussian tails in the dataset, we perform
an iterative fit, in which we calculate the standard deviation of the residuals to the fit ( $\sigma_{res}$ ) and discard any points lying more than $4\,\sigma_{res}$ from the fit. Of the 3760 fits, only 50 had any outliers and only 11 of these required more than one iteration to converge. Representative plots and fits are shown in Fig. 5.

In total, we have slope values for 3760 6-hr subsets. While the great majority of slope values are close to -1.0 and are normally distributed (see Fig. 6), a small number are implausibly large (up to +2035.9) or small (down to −102.6). Since we seek a representative slope value for each high/low day/night subset, we choose to limit the impact of these outliers by performing an iterative calcuation of the means. We calculate a mean and standard deviation, discard all slopes more than $3\sigma$ from the mean and iterate to convergence. Table 2 shows the results of these calculations, as well as sensitivity studies varying the length of the day and night intervals and the tightness of the cut in our iterative averaging.

## 4    The relationship between local $O_2$:$CO_2$ ratios and the local biotic exchange ratio (LBER)

In its simplest form, the variability in $O_2$ and $CO_2$ in the continental interior is an end-member mixing problem: Fossil fuel fluxes will give slopes ranging from -1.17 (coal) to -2.0 (methane) (Keeling, 1988; Steinbach et al., 2011), while biogenic fluxes will give slopes closer to -1.0. Thus, we expect air masses exposed to both types of fluxes will evolve with intermediate slopes.

Within this conceptual framework, a successful effort to extract LBER from the data depends on the relative sizes of the biogenic and anthropogenic fluxes. Only at times when photosynthesis and respiration dominate the land-air fluxes will observed slopes be close to LBER.

To test these assertions, we first use a Lagrangian transport model to estimate the region over which surface fluxes influence signals at our sampling site. We then use a simple 1-dimensional box model to establish the validity of the end-member mixing framework. Finally, we estimate the relative size of the biogenic and anthropogenic contributions.

### 4.1    The 6-hour provenance of atmospheric variability

We use a Lagrangian transport model to estimate the surface regions to which air parcels are exposed during the six hours prior to their arrival at the EMS tower. Specifically, we created back-trajectories for a range of dates and times, day and night, across seasons and years using the HYSPLIT Lagrangian transport model (NOAA, 2018) forced with NAM12 meteorology. We generated many sets of six 6-hour back-trajectories originating hourly during the daytime or nighttime intervals to determine the provenance of the air we analyze at Harvard Forest. An example of one such set of trajectories is shown in Fig. 7.

In general, the 6-hour provenance ranges from 50 km to 200 km, with little spatial variation over the 6-hour interval. Given the location of our study site, a $\sim 100$ km provenance, primarily from the west, means the air we analyze has travelled over a mostly forested landscape, occasionally passing small urban centers. This analysis is consistent with the work of Gerbig et al. (2006) who found a 10-fold (or greater) fall-off in the influence of fluxes more than $\sim 100$ km from the study site.

### 4.2    "Mixing" of slopes

To test the applicability of an end-member approach, we considered the simplest model with two contributors: generic fossil fuel, with an exchange ratio of -1.4, and biotic activity, with an exchange ratio of -1.05. These characterized the stoichiometry of surface fluxes used in a 1-dimensional box model, in which parcels of air travel forward in time in 12-minute steps, passing

over a fictitious landscape with forested, suburban and urban components. In 30 of these time steps the parcels exchange $CO_2$ and $O_2$ with the underlying sources and sinks, changing the composition of the parcel according to

$$C_{end} = C_{start} + \frac{F \cdot t}{h} \frac{22.4}{1000} \tag{6}$$

where $C$ is the mixing ratio of $O_2$ or $CO_2$ in $\mu mol\ mol^{-1}$, $F$ is the flux from the landscape underneath the box in $\mu mol\ m^{-2}\ s^{-1}$,

$t$ is the time step in seconds, $h$ is the height of the box in $m$, and the factors of 22.4 (the molar volume) and 1000 convert moles to liters and liters to $m^3$, respectively. For $h$ we use one half of the planetary boundary layer (PBL) height. We spin up the model for 24 hours and then record $O_2$ and $CO_2$ concentrations of consecutive parcels as they "arrive" in their final time steps over the same 6-hour day and night intervals used for the observations.

The $CO_2$ surface fluxes vary with time of day and day of year and are based on observations at Harvard Forest (Munger and Wofsy,

2017) and in urban (Bergeron and Strachan, 2011; Velasco and Roth, 2010; Ward et al., 2015) and suburban (Bergeron and Strachan, 2011) settings. The $O_2$ surface fluxes were derived from the $CO_2$ fluxes by assuming exchange ratios of 1.4 for fossil fuel combustion and 1.05 for biospheric activity. The PBL values are climatological hourly average values, calculated separately for summer and winter from the NAM12 dataset (NOAA, 2018). Within the STILT model (Gerbig et al., 2006), only air within the lower half of the PBL is influenced directly by surface fluxes. Air above this height is closely matched to the composition of

the free troposphere. Because we use the STILT model to infer the provenance of air we analyze, we define our box height as half the height of the PBL so that the two models (STILT and our 1-dim box model) are conceptually consistent. We performed sensitivity studies, varying the landscape composition (ranging from 100% forest to 100% urban), surface flux magnitude (values characteristic of winter and summer seasons) and PBL height (25-75% of the NAM12 values). In all cases, the $CO_2$ and $O_2$ values of the parcels entering the model were held constant, ensuring that any variability predicted by the model was a

result of the fluxes from the landscape over which the parcels travelled in the previous 6 hours and/or changes in PBL height.

We emphasize that the purpose of the 1-D model is to explore the validity of the end-member mixing approach. In particular, we ask whether a mix of sources will relate simply to a mix of slopes, despite the fluxes and the box height having their own diel cycles.

In all but a few special circumstances, the model predicts slopes that are completely consistent with the specified admixture

of the source fluxes over which the air parcels have traveled. Exceptions arise when the covariation of the boundary layer height and diel cycles in the fluxes conspire to bias slopes by a few percent. Results of various sensitivity tests and a discussion of these exceptions can be found in Table 5.1 of Conley (2018), along with a more complete description of the model and its forcing fluxes.

The 1-D model results give us confidence that slopes close to -1.0 are a closer representation of the LBER, while more

negative slopes have ever larger contributions from fossil fuel combustion. Though reassuring, we acknowledge the model results are not definitive due to the many simplifications involved. More sophisticated studies are needed to give a truly robust confirmation of the end-member mixing framework.

Aside from the slopes, the full range of $O_2$ and $CO_2$ variability predicted by the model is in the general range of our observations. Because of the crude nature of our model, this result is somewhat surprising. Our values for the box height and the molar volume are almost certainly wrong, so the data-model agreement may be coincidental.

## 5 Discussion of observations

Adopting the approach outlined above, we interpret slopes of our plots as averages of LBER (a number near -1.0) and external fossil-fuel influences (numbers ranges from -1.17 to -2.0), weighted by the relative contributions of these fluxes in the Harvard Forest "footprint."

Taking all data together, using 6-hour intervals, we find a slope of $-1.081 \pm 0.007$ (see Table 2). This value is a little higher than the oxidative ratio of organic material calculated by Worrall et al. (2013) and at the lower end of the range of possible values for the global exhange ratio quoted by Severinghaus (1995).

Ideally, our dataset would not include "poor" slopes – those that are weakly constrained by the data or unduly influenced by a small number of points. We found the most effective tool for excluding such slopes was a $3\sigma$ iterative cut. In practice, this cut only eliminates 6-hr intervals with slopes that are positive, or very negative ($< -2.0$). Inspection of a subset of the fits failing the cut shows they are indeed poor fits, as defined above. While the cut does not catch all poor slopes, we believe the slopes that are cut are poor ones.

We emphasize that this all-inclusive average slope is not a direct measurement of the LBER. Instead, it is a measure of the LBER mixed with the exchange ratio of fossil fuel combustion.

We also considered 2-week aggregation periods (instead of 6 hours). These slopes average $-1.19 \pm 0.02$. This is consistent with nearly equal contributions from fossil and biotic fluxes, as expected for an aggregation period with a provenance encompassing the entire continental United States (Gerbig et al., 2006).

### 5.1 Information from subsets of data

Considering particular subsets of the data (Table 2) may provide insight into the processes determining observed slope values. Thus, we categorize our 6-hour slopes by day and night, high and low, winter (Dec, Jan, Feb) and summer (Jun, Jul, Aug).

We begin with data aggregated in broad categories. Each group's average slope, calculated using the $3\sigma$ cut described above, is shown in Fig. 8. These groupings convey some unequivocal messages: Summer slopes are less negative than winter slopes. Day slopes are less negative than night slopes. Low-intake slopes are less negative than high-intake. Daytime data from the low intake are least negative of all.

We also consider narrower categories, each with a unique set of slopes. The average slopes for each category are shown in Fig. 9. For these averages, we impose a $2.5\sigma$ iterative cut because a $3\sigma$ cut fails to reject several unphysical fits (with slopes $> 2.0$) in the winter subsets, badly biasing the mean.

The most striking feature of this plot is the large scatter in the winter data. Even with the $2.5\sigma$ iterative cut, the slopes within a category range widely and are difficult to interpret. In contrast, the summer data comprise much tighter groups and have well-defined mean values, showing the same day/night and high/low relationships as the full dataset.

Explaining these patterns is challenging. Possible factors determining the observed slope include the magnitudes of the fluxes, the dynamics of the atmosphere, and variation of the biospheric exchange ratio over time.

### 5.1.1 The relative size of fluxes

The relative weighting for combustion and biogenic influence in the slope will depend in part on the magnitudes of the respective fluxes across the region. Our best estimate of combustion fluxes comes from Sargent et al. (2018), who use an inverse analysis of $CO_2$ concentration data to infer a value of $0.8\,\mathrm{kg\,C\,m^{-2}\,y^{-1}}$ (i.e. $2.4\,\mu\mathrm{mol\,C\,m^{-2}\,s^{-1}}$) for Boston and its suburbs. We can relate this to Harvard Forest using the ACES high-resolution emission inventory developed specifically for the Northeastern US (Gately and Hutyra, 2017). ACES shows a strong decrease in anthropogenic emissions moving from Boston to Harvard Forest and beyond to the west and north, with the area around the tower showing annual mean emissions closer to $0.3\,\mu\mathrm{mol\,C\,m^{-2}\,s^{-1}}$.

For the biogenic fluxes, we use eddy-flux measurements of net ecosystem exchange (NEE) at Harvard Forest (Munger and Wofsy, 2017). In the summer, the daytime mean NEE at Harvard Forest is of order 10 or 20 $\mu\mathrm{mol\,m^{-2}\,s^{-1}}$ for coniferous and deciduous forest, respectively, and around $5\,\mu\mathrm{mol\,m^{-2}\,s^{-1}}$ at night. Winter NEE averages range from 2 to 3 $\mu\mathrm{mol\,m^{-2}\,s^{-1}}$.

Collectively, these data show the regionally averaged biogenic influence on $CO_2$ (and by extension $O_2$) is substantially larger than the fossil fuel source during the growing season, and dwarfs it during daytime. In winter months, anthropogenic $CO_2$ emissions have, at most, a magnitude comparable to the biogenic flux. Further evidence that combustion sources are a minor player at Harvard Forest during the growing season comes from the observation by Potosnak et al. (1999) that $CO_2$ and CO are correlated in winter but not summer. Finally, we note that the modeling work of Gerbig et al. (2006) shows short-term variability in $CO_2$ is dominated by the predicted influence of vegetation.

While US EPA's inventory (Agency, 2017) shows that the primary upwind sector (west-northwest) (Moody et al., 1998) is essentially free of power plants for 100km or more, because we lack contemporaneous measurements of CO (a tracer of combustion), it remains a possibility that plumes of power plant exhaust might occasionally influence our measurements.

In summary, based on the estimated magnitudes of the biotic and combustion fluxes, we expect the summer daytime observations of the $O_2$:$CO_2$ slopes to be most strongly linked to the LBER. Furthermore, the link between winter variability in $CO_2$ and CO observed by Potosnak et al. (1999) offers one possible explanation for the much greater scatter we see in our winter data.

Comparing day and night data, fossil fuel influence may be more apparent at night when biotic fluxes are reduced (Munger and Wofsy, 2017) but fossil fuel combustion falls only slightly (Bergeron and Strachan, 2011; Velasco and Roth, 2010; Ward et al., 2015).

### 5.1.2 Atmospheric Dynamics

With the diel cycle of insolation, the planetary boundary layer cycles between nocturnal stability and diurnal convective over-turning. This is clearly evident in the climatological diel cycle of $u^\star$ at Harvard Forest (Munger and Wofsy, 2017). Based on these data, we defined our 6-hour daytime period to capture full photosynthetic activity and maximal vertical mixing, while our night interval captures pure respiration and maximal stratification. These dynamics suggest that air at both of our intakes would be closer in composition to air aloft during the middle of the day. Thus, daytime slopes might show more influence from distant fluxes including more fossil fuel combustion (and thus, be more negative).

In addition, we expect nocturnal stability to trap fluxes from soil respiration close to the ground, amplifying their impact, particularly on the lower intake (Munger, 1996; Urbanski et al., 2007). This will lead to biologically dominated nocturnal slopes. Assuming the respiration-driven slope is closer to -1.0 than fossil fuel, we would expect to find the least negative slopes at night.

The fact that we see less negative slopes during the day than at night ($-1.02 \pm 0.01$ vs. $-1.12 \pm 0.01$) suggests that either other factors are more important (Sec. 5.1.1 and Sec. 5.1.3) or this picture of atmospheric motion is oversimplified.

Another implication of diurnal convection and nocturnal stratification would be similar behaviour of the high and low intakes during daylight hours and greater differences at night. We do see a suggestion of this in our summer data, when the high-low difference appears to be smaller during the day ($-0.015 \pm 0.02$ vs. $-0.03 \pm 0.01$), but uncertainties are substantial. We also expect any differences driven by stratification to be reduced during the winter. Unfortunately, our winter data are too variable to meaningfully test this expectation ($-0.1 \pm 0.1$ and $-0.16 \pm 0.08$).

Yet another possibility is a daytime temperature inversion within the forest canopy. This arises because air is warmed from above by incoming sunlight striking leaves (Jacobs et al., 1992). However, profiles of $CO_2$ and $O_2$ on the tower show that at Harvard Forest these inversions are often eroded by gusts from above penetrating the canopy and usually only persist in the first meter or two above ground level.

Regardless of time of day or year, velocities within the canopy will be frictionally reduced. This might suggest an enhancement of local signals (presumably with less fossil influence and thus, less negative slopes) on the lower intake. This is very much consistent with our observations: With the $3\sigma$ cut, low and high slopes are $-1.04 \pm 0.01$ and $-1.11 \pm 0.01$, respectively. Results are very similar with the $2.5\sigma$ cut.

Finally, we note that $u^\star$ varies seasonally as well as daily. Not only is $u^\star$ roughly twice as high in the winter as the summer, but it is far more variable. This may explain why the slopes in the winter months exhibit so much variation.

### 5.1.3 Possible variation in the biotic exchange ratio

Another possible driver of variability in slopes is variation in the biotic exchange ratio over time. For example, nocturnal plant respiration might consume molecules richer in nitrogen with higher oxidative ratios than the molecules being produced during photosynthesis (Severinghaus, 1995). Even in the complete absence of dynamical influences or fossil fuel contamination, this would lead to more negative slopes at night.

This scenario is built on a violation of mass balance that cannot persist indefinitely. Eventually, the nitrogen-poor material that was synthesized above ground must also be respired, changing atmospheric $O_2$ and $CO_2$ with slopes closer to -1.0.

The viability of this explanation for our more negative nocturnal slopes becomes a question of equilibrium. In illustrative (and highly simplified) terms: Is the forest as a whole out of balance, engaged in a long-term accumulation of nitrogen-poor, refractory trunk wood and a steady diversion of nitrogen into labile leaves and rootlets? Or is the forest in balance, with as many tree trunks being decomposed as are being synthesized?

Evidence pointing to a balanced forest are the isotopic measurements by Wehr and Saleska (2015) and Wehr et al. (2016), which suggest much of Harvard Forest's respired carbon was assimilated no more than a week or two earlier. On the other hand, Trumbore (2000) used radiocarbon measurements to show that the mean age of respired soil carbon is 3 years. These two studies are not entirely incompatible if Trumbore's mean age is comprised of, for example, 60% very young carbon with the balance coming from much older pools.

At this point, we are not in a position to say whether the day-night difference in slopes is due to a forest that is out of equilibrium, or whether we are measuring respiration of the full range of substrates and something else is responsible for the day-night difference in slopes.

## 5.2 Temporal variability in slopes

Our data are sufficiently abundant that we can examine the values of slopes as a function of time on both interannual and seasonal time scales. Figure 10 shows that one year looks much like the next, but stacked years reveal a seasonal variation in the slopes.

The seasonal cycle in slopes may be due to seasonal changes in LBER itself. Trees synthesize many different compounds throughout the year (Seibt et al., 2004), each with their own oxidative ratios (Bloom et al., 1989). However, direct measurements of leaves and tree-rings indicate oxidative ratios in these tissues appear invariant through the year (Gallagher et al., 2017).

On the other hand, the total amount of biological activity unquestionably varies seasonally, while the fossil fuel contribution remains roughly constant. As mentioned above, earlier work at Harvard Forest using carbon monoxide and acetylene (Potosnak et al., 1999) shows clearly that fossil fuel combustion is a driver of $CO_2$ variability in the winter, but is overwhelmed by biotic signals during the summer. Thus, the simple seasonal shift in the biotic/fossil-fuel balance is a likely explanation for some, if not all, of the seasonal cycle in slopes.

Spectral analysis (using Lomb-Scargle periodograms to accommodate our irregularly spaced data (Press et al., 2007)) of various subsets of the data shows some evidence of the seasonal cycle mentioned above. It appears to be mostly in the night/high-intake subset. There is also a suggestion of multi-year variation (0.6 cycles/year) in the day-low dataset and 3-week periodicity in the day-high and night-low data. Identical spectral analyses of environmental variables measured at Harvard Forest (temperature, net ecosystem exchange, ecosystem respiration, photosynthetically active radiation and $u^\star$) all show highly significant seasonal and diel cycles, as expected.

The three-week cycle in the day-high and night-low records is perplexing. No such cycle appears in the environmental variables listed above. Thus, our working hypothesis is that the 3-week cycles are an experimental artifact resulting from the timing of runs of calibration tanks and swapping of working tanks.

### 5.3 Selecting periods of heightened biogenic signal

Given the competition between biogenic and anthropogenic influences in our observations, we searched for periods when the biogenic contribution might be most dominant. We selected times when the fluxes in the immediate neighborhood were likely to dominate. Our assumption was that background variation (i.e. compositional variability due to influences beyond the $\sim 100$ km provenance of our measurements) is likely to carry signals of continental-scale gradients with more substantial fossil fuel influences embedded therein.

One approach we considered was based on $u^\star$. We hypothesized that slopes calculated when $u^\star$ was low were more likely to be biogenic, since the air at our intakes was exchanging minimally with the overlying boundary layer. However, a scatter plot of slopes as a function of $u^\star$ showed no correlation. Similarly, slope values were insensitive to discrete cuts on average $u^\star$ ($< 40$ cm s$^{-1}$), maximum $u^\star$ ($< 50$ cm s$^{-1}$) and the scatter in $u^\star$ ($\sigma_{u^\star} < 10$ cm s$^{-1}$).

We also explored the possibility of a link between slopes and the ranges of $CO_2$ and $O_2$ during the 6-hour periods. Again, 15  no correlation was apparent, whether with discrete cuts during summer and winter periods separately, or with scatter plots of slope vs. range.

We tested the hypothesis that summer periods of extended high barometric pressure with languid anticyclonic circulation might be dominated by local fluxes. Again, we observed no correlation between slope and barometric pressure, whether with discrete cuts or with scatter plots.

Finally, we looked for correlations between slopes and wind speed, direction and variability. We found none, even when winds were from the relatively polluted sector to the southwest.

In short, relatively simple indicators of reduced advective transport show no correlation with shallower slopes. This insentivity of our result adds support for our model of a geographically modest footprint comprised of a landscape in which biogenic fluxes are much stronger than anthropogenic ones.

## 6  Conclusions

Having measured changes in atmospheric $CO_2$ and $O_2$ in a mixed deciduous, mid-latitude forest with a precision of a few $\mu$mol mol$^{-1}$, we find that these species covary with an all-data average molar ratio over 6-hr periods of $-1.081 \pm 0.007$ ($O_2 : CO_2$). In the absence of fossil-fuel influences, our measured ratios reflect the local signature of photosynthesis and respiration (LBER). If our data represent a mix of forest and fossil fuel fluxes, -1.081 is a lower (most negative) limit on 30  LBER. Adopting a two-end-member mixing model, the best value would be taken from the subset of the data with the smallest fossil fuel contribution and presumably, the shallowest measured slope. Restricting our dataset to daytime values from the low

intake in summer (when we expect local influences to be maximal due to vigorous biotic activity and modest coupling with the air above the canopy), we find a slope of $-1.03 \pm 0.01$.

Our values are consistent with measurements in other temperate forests. The consistency of elemental ratios in the work of Worrall et al. (2013) across a range of study sites suggests the Harvard Forest result may have broad applicability. Furthermore, basic constraints in plant physiology argue against wildly divergent values across species (Björkman and Demmig, 1987). Nonetheless, similar studies at other sites are required before concluding a value very close to 1.0 is globally representative.

Also, it is still possible that gross photosynthetic and respiratory exchanges could exhibit $O_2 : CO_2$ ratios individually that differ from the (much smaller) net uptake of $CO_2$ and release of $O_2$. For this reason it is important to track $O_2$ in process-based ecosystem models, enabling predictions of both LBER and $\alpha_B$ while matching observations.

Even with this qualification, our measurements provide valuable insight into the nature of $O_2$ and $CO_2$ exchange in a temperate forest. They also suggest that the best value for $\alpha_B$ in calculations partitioning global carbon fluxes may be lower than 1.1, the value widely used to date (Keeling and Manning, 2014). At the least, studies that rely on $\alpha_B$ should explore sensitivity to values as low as 1.0.

If our results do indeed match the ratio of the global net carbon sink, the use of 1.03 for $\alpha_B$ leads to a 7% increase in terrestrial carbon uptake and an equivalent reduction in oceanic carbon storage (see Eq. 8 of Keeling and Manning (2014)). While this adjustment is within the uncertrainties on these terms in the global carbon budget, it would nonetheless be an important correction.

*Data availability.* Battle M, Munger W. 2018. Atmospheric Oxygen and Carbon Dioxide at Harvard Forest EMS Tower since 2006. Harvard Forest Data Archive: HF306.

*Author contributions.* MB conceived of the experiment. MB built and operated the hardware and wrote software with assistance from JWM, ES, RP, RH, ZD, JS, JW, KG, SS, JC and SD. MB and MC performed data analysis. MB wrote the manuscript with assistance from JWM and MC.

*Competing interests.* The authors have no competing interests to declare.

*Acknowledgements.* Analysis of our data was greatly aided by a suggestion from Alessandro Cescatti. We also thank Maryann Sargent and Richard Wehr for helpful conversations and Taylor Jones for sharing code. Robert Stevens helped with machining, Dj Merrill provided computing support and Emery Boose and John Budney gave invaluable assistance at Harvard Forest.

We thank Britton Stephens and an anonymous reviewer for exceptionally thoughtful reviews and suggestions that greatly improved the manuscript.

Operation of the Harvard Forest flux tower site is supported by the AmeriFlux Management Project with funding by the U.S. Department of Energy's Office of Science under Contract No. DE-AC02-05CH11231, and additionally as a part of the Harvard Forest LTER site supported by the National Science Foundation (DEB-1237491). Support for the tower operation during the period examined here, as well as partial support for the $O_2$- $CO_2$ sampling equipment, came from the U. S. Dept of Energy Office of Science through programs that are no longer active (including the National Institute of Global Environmental Change and the National Institute for Climate Change Research). Further support for the sampling equipment was provided by Bowdoin College.

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

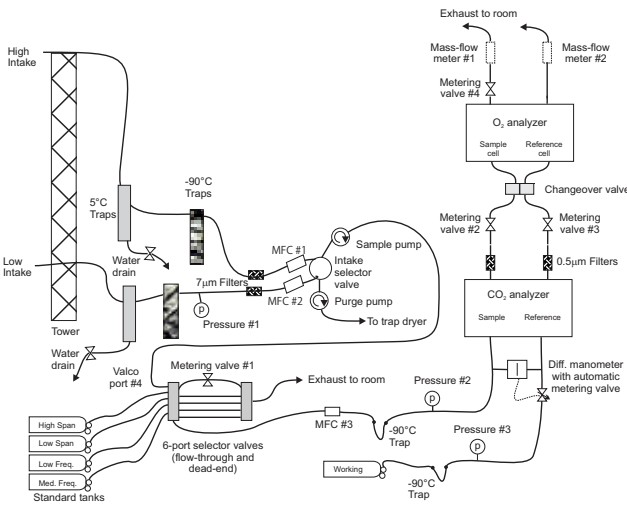

**Figure 1.** Schematic diagram of the instrumentation installed at Harvard Forest.

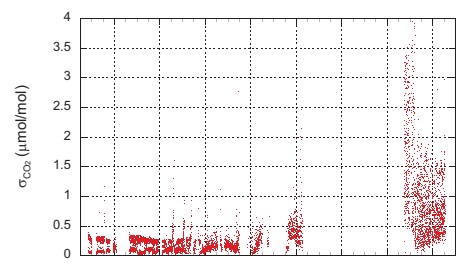

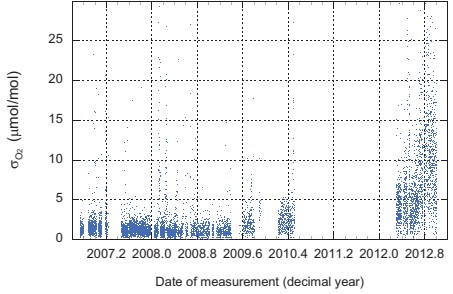

**Figure 2.** The scatter (standard deviation from the mean) of routine $CO_2$ and $O_2$ measurements of calibration tanks. Values plotted here apply to individual $CO_2$ measurements (1 Hz) and single difference-of-difference $O_2$ values (1/48s).

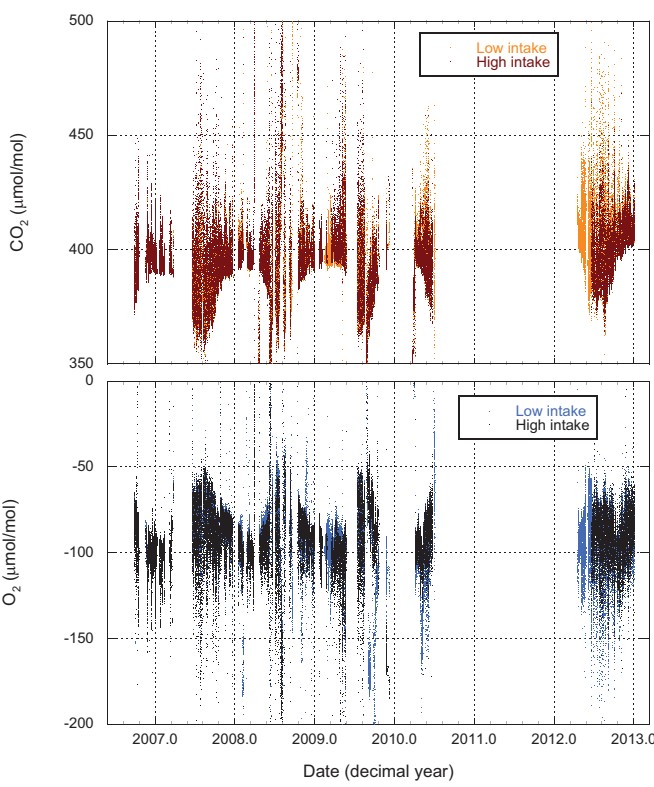

**Figure 3.** $CO_2$ and $O_2$ measured from both high and low intakes at Harvard Forest. The data have been processed (Sec. 2.2) and are presented on the WMO and S2 scales. Each point (spaced 11 min apart) represents the average atmospheric composition for a 4.8 min period.

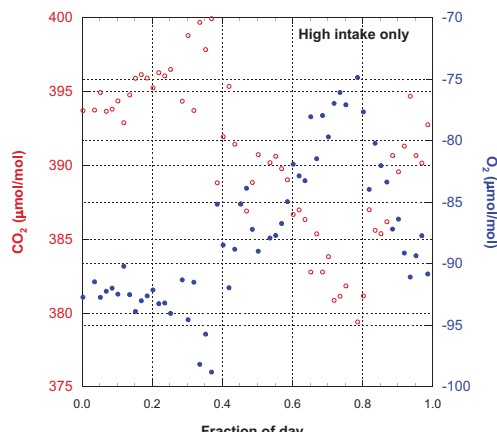

**Figure 4.** A detail from Fig. 3: $CO_2$ and $O_2$ measured from the high intake on Sept. 29, 2006. This is simply one example of the diel cycle of these species. Note the strict inverse variation in $O_2$ and $CO_2$, even when there are large deviations from a smooth temporal evolution.

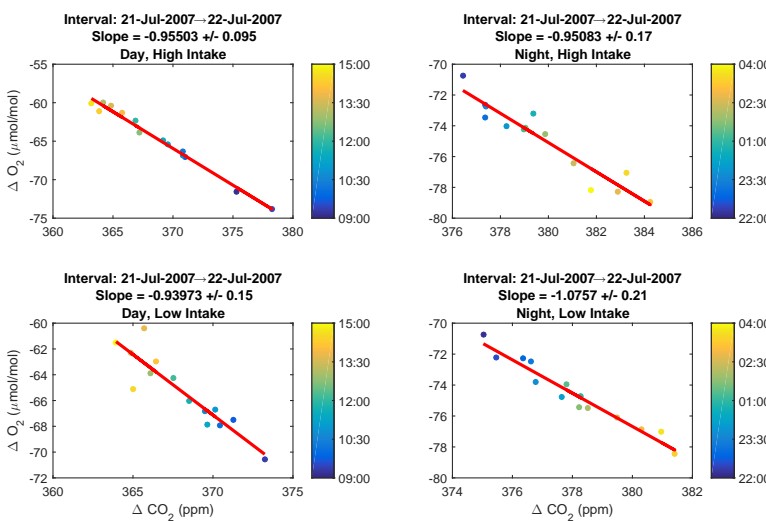

**Figure 5.** Representative examples of slope-plots: data collected during day and night periods from the high and low intakes on July 21-22, 2007. The colors of the points indicate the time of collection within the 6-hr intervals, as keyed in the color bar to the right of each plot. Errors on the slopes are purely statistical.

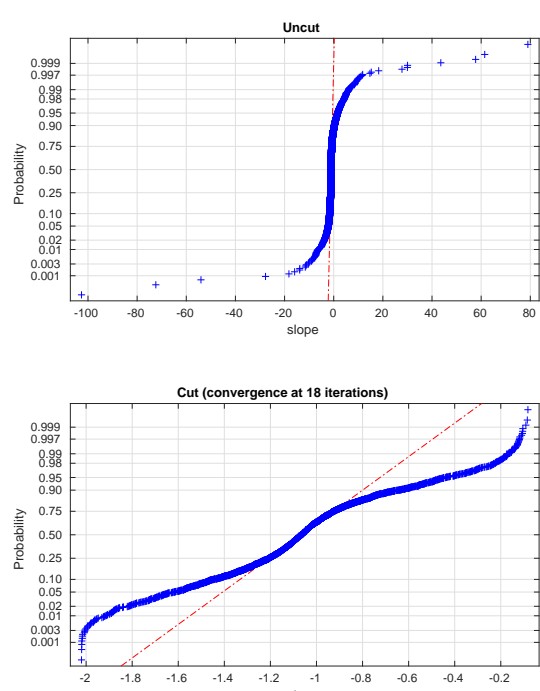

**Figure 6.** Probability plots of slopes for the entire dataset, before and after a $3\sigma$ iterative cut. The dash-dot lines indicate the shape expected for a purely Gaussian distribution. In the interests of clarity, we omit five points with extremely large or small slopes from the upper panel. See Table 2 for statistical details of these distributions.

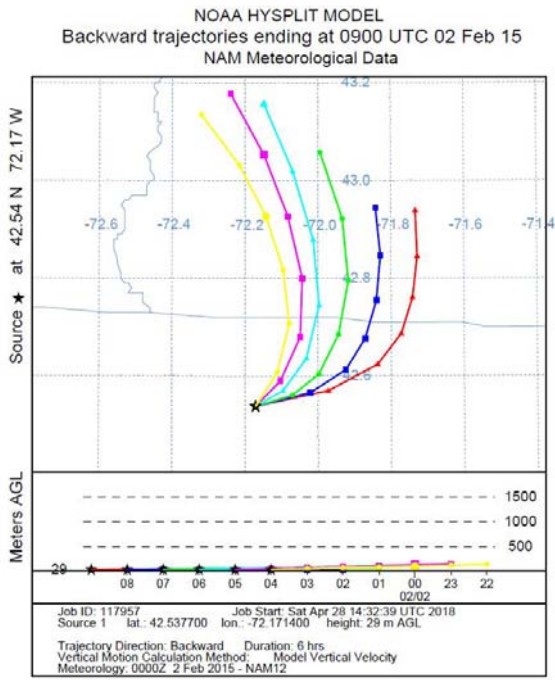

**Figure 7.** An example of 6-hour back-trajectories for Harvard Forest on Feb. 2, 2015. Trajectories start hourly between 10pm and 4am EST, corresponding to the nighttime interval for slope plots. State lines for New Hampshire, Vermont and Massachusetts are in gray. On this particular night, the source region was fairly constant, so changes in $O_2$ and $CO_2$ are relatively likely to be primarily due to local influences.

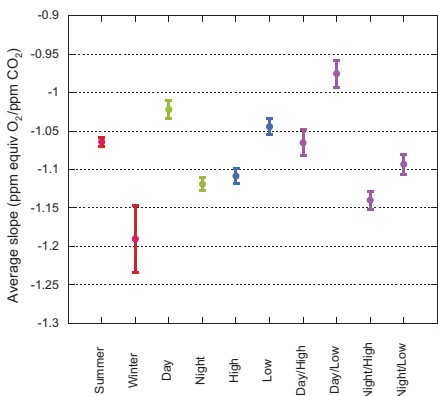

**Figure 8.** Average slopes for various subsets of the data. Error bars show the standard error on the mean. Subsets within each group (i.e. sharing a color) are non-overlapping.

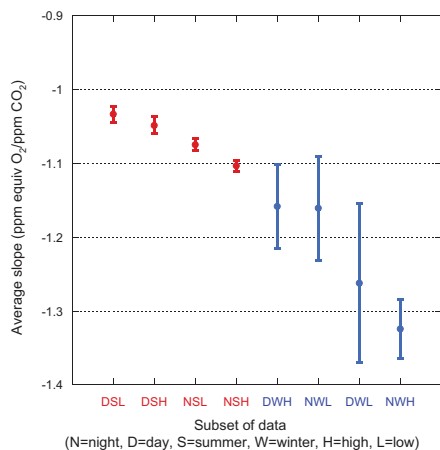

**Figure 9.** Average slopes for various non-overlapping subsets of the data. Error bars show the standard error on the mean. Winter data are shown in blue; summer in red. Categories are ordered by increasingly negative mean values for convenience.

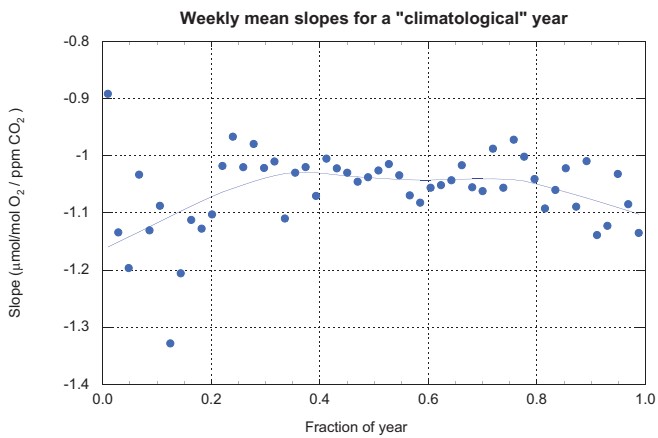

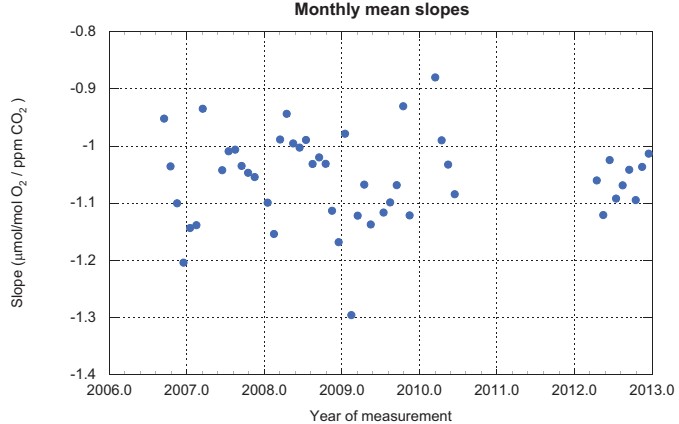

**Figure 10.** Upper panel: Slopes (averaged weekly) for a climatological year. Lower panel: Slopes (averaged monthly) for the entire period of study. The gray line in the upper panel is a locally weighted least-squares fit with 46% smoothing. It is simply intended as an objective guide to the eye.

**Table 1.** Nominal composition of calibration tanks. Actual values vary slightly as cylinders are used and refilled. The $O_2$ values quoted here were measured in the lab of R. Keeling, while the $CO_2$ values were measured by the CCGG lab of NOAA-ESRL-GMD. Based on replicate measurements, the $O_2$ values are typically determined to within roughly $\pm 1.5$per meg on the S2 scale, while the $CO_2$ values on the WMO X2007 scale are known to about $\pm 0.2\ \mu\mathrm{mol\ mol}^{-1}$ (Hall, 2017). The cylinders are Luxfer N265 uncoated aluminum, with 51-14B regulators from Scott Specialty Gases (now Air Liquide). Tanks and regulators are stored horizontally in an insulated box to minimize thermal and gravitational fractionation.

| Tank name | $O_2$ (per meg on S2 scale) | $CO_2$ ( $\mu\mathrm{mol\ mol}^{-1}$ ) | Analysis frequency |
|---|---|---|---|
| High Span (HS) | +125 | 350 | 4/day |
| Low Span (LS) | -850 | 525 | 4/day |
| Medium Freq. (MF) | -360 | 375 | 1/3 days |
| Low Freq. (LF) | -470 | 390 | 1/week |

**Table 2.** Table of average values for all slopes and four data subsets. $\sigma$ is 1 standard deviation, SEM is the standard error of the mean ($\sigma/\sqrt{n}$), and $n$ is the number of slope values surviving the iterative cut.

| Interval length and cut | Data set | mean | $\sigma$ | SEM | $n$ |
|---|---|---|---|---|---|
| 6-hour ($3\sigma$) | All data | -1.081 | 0.38 | 0.007 | 2950 |
| | Day, high intake | -1.07 | 0.46 | 0.02 | 774 |
| | Day, low intake | -0.98 | 0.46 | 0.02 | 684 |
| | Night, high intake | -1.14 | 0.33 | 0.01 | 801 |
| | Night, low intake | -1.09 | 0.36 | 0.01 | 725 |
| 4-hour ($3\sigma$) | All data | -1.071 | 0.66 | 0.008 | 3070 |
| | Day, high intake | -1.04 | 0.58 | 0.02 | 751 |
| | Day, low intake | -0.94 | 0.66 | 0.03 | 679 |
| | Night, high intake | -1.15 | 0.34 | 0.01 | 808 |
| | Night, low intake | -1.10 | 0.37 | 0.01 | 725 |
| 6-hour ($4\sigma$) | All data | -1.005 | 0.77 | 0.01 | 3347 |
| | Day, high intake | -1.03 | 0.95 | 0.03 | 877 |
| | Day, low intake | -0.85 | 0.92 | 0.03 | 782 |
| | Night, high intake | -1.11 | 0.63 | 0.02 | 898 |
| | Night, low intake | -1.01 | 0.71 | 0.02 | 833 |
| 6-hour (no cut) | All data | 5.93 | 205 | 3 | 3760 |
| | 6 outliers cut | **31** -0.05 | 37 | 0.6 | 3754 |