# Peer review of "Atmospheric measurements of the terrestrial O2:CO2 exchange ratio of a mid-latitude forest"

_Atmospheric Chemistry and Physics, 2018_

## Referee Comment (RC1) · Anonymous Referee #1 · 13 Dec 2018

In this paper, Battle and co-authors, conducting precise observation atmospheric CO2 and O2 concentrations at the Harvard Forest during 2006-2013, examine the correlative changes to evaluate the O2:CO2 exchange ration associated with terrestrial biospheric processes, $\alpha$B. Although the value of $\alpha$B is basic parameter to understand the atmospheric O2 variation and the value of 1.10 is commonly used for long time, there are still discussions regarding the absolute value. Finding the average correlation slope in the forest is significantly lower than 1.1, the authors conclude that the value of 1.1 for $\alpha$B should be adjusted to slightly lower value as several other studies have already suggested. I believe this study considerably contributes to the atmospheric O2 study field and contains material that should be published in Atmospheric Chemistry and

[Figure]

Physics. However, I find a problem in processing the observed data and incomplete descriptions of the modeling studies to evaluate the contributions from the biotic and combustion emissions. An effort should be made by the authors to improve the data treatment and clarity of the manuscript before acceptance.

General comments: Although the authors carried out highly precise measurements of the atmospheric O2, I have a concern about the units of the O2 measurements used in this paper. The authors use units of mole fraction, $\mu$mol per mol, to express the O2 variations as well as CO2. However, the use of mole fraction to express variations in major atmospheric constituents like O2 are very confusing because of the influence of dilution effect (e. g. Keeling et al., 1998). For example, adding 1 $\mu$mol of CO2 to and removing 1.1 $\mu$mol of O2 from an air parcel containing 1 mol of dry air with 0.21 mol of O2 results in a 1 ppm increase in the CO2 mole fraction and 1.08 ppm decrease in the O2 mole fraction. Therefore, the correlation slope based on the atmospheric observation is -1.08, which is lower than the original O2:CO2 ratio of 1.1. In the similar way, if the authors truly use the mole fractions both for O2 and CO2, the correlation slope of 1.10 corresponds to the O2:CO2 exchange ratio of 1.126. To avoid such confusing situation, units of per meg was defined by Keeling and Shertz (1992). The authors, therefore, should use "per meg" or ppm equivalent calculated as a product of "per meg" and 0.2094.

To evaluate the value of $\alpha$B from the field observation in the forested area, it is critically important to quantitatively assess influences of emissions from the fossil fuel combustions. The authors examine the contributions from the fossil fuel and biogenic fluxes in Section 4 by using a Lagrangian transport model. However, the results of the examinations are not clearly shown as are described in detail in specific comments as follows. In addition, except for such a modeling approach, I think that the observations like CO and 14C are useful to assess the influence of the fossil fuel fluxes. If such observations were carried out at the site, it would be better to use those data.

Specific comments: Page 1, line 7: It says here that the upper intake is placed about 6

m above the forest canopy. But, the height is about 5m in Method (page 3, line 4).

Page 1, line19 and 20: It would be better to number the equations. In addition, I understand that these budget equations were original forms given by Keeling and Shertz (1992), but recent studies generally include ocean $O_2$ outgassing term in the $O_2$ budget equation. So, I think it would be better to add the $O_2$ outgassing term because it's considered that its uncertainty is the largest source of error for the carbon budget estimation based on the atmospheric $O_2$ measurements.

Page 3, line 3: I think there is no need to show the position (latitude and longitude) of the site with a 0.1 m precision.

Page 3, line 21-23: It would be better to mention the size and the material of the calibration tanks. What are the uncertainties of the concentrations of $O_2$ and $CO_2$ of the standard cylinders listed in Table 1. Were the $O_2$ concentrations in the calibration tanks stable during the entire observation? Could you show any experimental evidences of the $O_2$ stability? In addition, "per meg" units are used for the $O_2$ concentrations in Table 1, but there is no explanation of the units in the text.

Page 5, line 30: Does the "intake selector valve" correspond to the "Cross-over ball valve" in Figure 1? It should be clarified.

Page 6, line 7-10: I think that the long-term stability in the $O_2$ and $CO_2$ concentrations in the reference tanks is also critically important to accurately assess the $O_2$:$CO_2$ exchange ratio. Thus, it would be better to show some experimental results to confirm the stability of the reference tanks.

Page 6, line 27: During the 6-year duration, the $O_2$ concentration of the background air decrease by more than 20 ppm. But it is difficult to see such decreasing trend in the $O_2$ time series shown in Fig. 3.

Page 6 line 27: It would be better to change "due to fossil fuel combustion" to "mainly due to . . .".

Page 7, line 29: It is mentioned that in the 1-deminsional box model the air parcels travel forward in 12-minute steps. I think that the air parcels move along with the backward trajectories computed by a Lagrangian transport mode. Is it right? If so, the authors should clarify that.

Page 8, line 3-4: Here, the authors mention that the PBL height used in atmospheric transport models like STILT is used for h. However, in the following section, the authors mention that climatological hourly averages from the NAM12 dataset are used as the PBL values. Are these PBL heights same? Please clarify it.

Page 8, line 11-12: I don't really know what the sentence, "We performed sensitivity . . . and PBL height", means. How are the landscape composition, surface flux magnitude, and PBL height varied? Are they varied within the uncertainties? What is the ranges of the variations? Please clarify it.

Page 8, line 16-17: Here the authors mention that the results can be found in Conly (2018). But, Conly (2018) is a bachelor's thesis, not a peer-reviewed paper. So, I think the authors should show at least main results of the 1-D model experiments here.

Page 8, Section 4.2 (Region of influence): Why don't the authors show the average footprint of the atmospheric observation at the EMS tower? Would comparison between the average footprint and the maps showing biogenic and fossil fuel fluxes convince the

Page 8-9: Why don't the authors quantitatively evaluate the fossil fuel-derived $CO_2$ contributions by using the above mentioned 1-D box model and the $CO_2$ biogenic and fossil fuel-derived fluxes described in Section 4.3? I believe that such an approach is simple and straightforward.

Page 9, line 4-5: How do the authors obtain the number "10-20 times"?

Page 10, line 2-3: Can the large $\alpha$B value for the soil respiration explain the high/low difference of the observed $-O_2/CO_2$ exchange ratio?

Page 10, section 5.3 (Temporal variability in slopes): Since the seasonal variation in the slope and $\alpha$B is discussed in this section, the seasonal O2/CO2 slopes should be shown. The authors mention "The seasonal cycle in slopes may be due to seasonal changes in $\alpha$B itself" (page 10, line 32). But immediately after that, it's mentioned that the fossil fuel contribution has significant seasonal variability: strong in winter and weak in summer. Do the authors consider that $\alpha$B changes seasonally? If so, seasonal variation in $\alpha$B estimated from the experimental result of this study should be clarified because it is very important to understand the atmospheric O2 and APO variations.

Page 17, Figure 1: Were the standard tanks placed vertically on the floor as shown in the figure? The standard tanks used for the O2 measurements are usually placed horizontally in the thermally insulated box to reduce fractionation effect on the O2 concentration due to the gradients of temperature and hydrostatic pressure within the tanks.

Page 22, Figure 6: What are the dashed-dotted lines in the figures? There is no explanation in the text and the figure captions.

Page 23, Figure 7: Why the backward trajectories after the observation period are adopted in the figure?

Page 24, Figure 8: These histograms are not discussed in the text.

Page 27, Table 2: Why is the order of the four data subsets for the 4-hour interval length (night-h, night-l, day-h, day-l) different from that for the 6-hour interval length (day-h, day-l, night-h, night-l)?

Please also note the supplement to this comment:
https://www.atmos-chem-phys-discuss.net/acp-2018-1041/acp-2018-1041-RC1-supplement.pdf

---

## Referee Comment (RC2) · Stephens (Referee) · 20 Dec 2018

Review of Battle et al. "Atmospheric measurements of the terrestrial $O_2$:$CO_2$ exchange ratio of a mid-latitude forest."

Britton Stephens, National Center for Atmospheric Research, Boulder, CO, USA

**Overview:**

This is a fantastic and much needed data set and the authors have done a careful job on the analyses. What I find particularly compelling are the consistent and systematic differences in $O_2$:$CO_2$ ratios detected at night versus day, and within vs above the canopy, which likely will be of great use to ecosystem modelers tuning a new generation of ecosystem models that track $O_2$ and which are needed for investigating controls on $\alpha_B$. This study is deserving of publication in ACP, and overall the presentation is good. However, I disagree with the broad extrapolation of the results to $\alpha_B$ and have several other substantive comments that I hope the authors are able to address.

**Major comments:**

1) Extrapolation to $\alpha_B$

The parameter $\alpha_B$ in the global carbon budgeting exercise corresponds to the ratio of the global net $O_2$ and $CO_2$ exchanges associated with the *unknown* net land carbon sink and any other net land oxygen sources. Because it is the ratio of unknown processes, it is unknowable itself, and all we can do is refine our best guesses of it based on measurements of various pools and fluxes, such as those as presented here. The misrepresentation of measurements of ratios of specific pools or fluxes as measurements constraining the global net $\alpha_B$ already exists in the literature, so the authors are not the first to make this generalization, but I do not think it is justified. Instead, they should state explicitly what is measured and clearly indicate the assumptions that are involved in extrapolation to a global ratio of the net land sink.

More specifically, the measured ratios presented here represent either the local respiratory flux (night), the combined influence of local respiration and photosynthesis (day), or the combined influence of these processes and external (e.g. fossil fuel) influences. They do not represent the $O_2$:$CO_2$ ratio of the net carbon sink and oxygen source in Harvard Forest, nor of course globally. Indeed, on local scales if photosynthesis happens at a slightly different ratio than respiration and these two components are nearly in balance, any value (from – to + infinity) is possible for the ratio of the net exchanges. On global scales, we know the net carbon sink is far from zero, so its ratio to the net $O_2$ source is constrained to be close to that of the corresponding pool(s). However, if the net sink is a result of more leaves, more fine roots, more soil organic matter, or more stem wood, we would expect very different ratios locally, and values for $\alpha_B$ globally.

Especially because this paper identifies clear differences between day and night ratios, possibly indicating systematic differences between photosynthesis and respiration, it is important to be very careful when using either of these measurements, or their average, to estimate a ratio of net fluxes, locally or globally.

Fixing this oversimplification is largely a matter of wordsmithing, and I make specific suggestions here. However, the authors may prefer to use additional terms (e.g. photosynthetic quotient, respiratory quotient, or oxidative ratio) or to define new ones to explicitly describe what has been measured.

1.1) Specific suggestions to more accurately discuss relationships to $\alpha_B$

Page 1 Lines 4-5, change "terrestrial photosynthesis and respiration" to "the net land sink"

Page 1 Lines 10-11, delete ", suggesting that this slope is our best estimate of $\alpha_B$"

Page 1 Line 12, delete the last clause or change to something along the lines of "a more complete picture of the ratios for the component fluxes and potential pools for the net sink is needed." Also, it would be good to call here or in the conclusion for ecosystem modeling that tracks $O_2$ and can explore potential values of $\alpha_B$ while matching observations.

Page 2 Line 2, change "terrestrial organic matter" to "the global net land carbon sink"

Page 2 Line 7, replace "effectively $\alpha_B$" with "one contribution to $\alpha_B$"

Page 2 Line 13, change "an average" to "the net imbalance"

Page 2 Line 17, add "of $\alpha_B$" after "estimates"

Page 2 Line 20, change "An alternative approach" to something like "Another method that can inform on the processes responsible for $\alpha_B$"

Page 2 Lines 23, change "a whole-ecosystem average value of $\alpha_B$" to "the ratio for a particular flux component or mixture of components at a particular time for the ecosystem."

Page 2 Line 25, add "contributions to" before "$\alpha_B$"

Page 2 Line 31, change "land-ocean partition" to "ecosystem and process specific ratios"

Page 7 Line 17, change section heading to something like "the relationship between local $O_2$:$CO_2$ ratios and $\alpha_B$"

Page 9 Line 9, change "$\alpha_{FF}$" to "external fossil-fuel (numbers from 1.17 to 2.0) influences"

Page 9 Lines 12-14, change "in agreement with the values" to "similar to the stock-based estimate" and add "but is not a direct measure of the ratio of the net carbon sink at Harvard Forest or globally" to the end of the paragraph.

Page 10 Lines 4-6, we know the forest is not in balance, especially during summer but also on annual means, so this should not be considered just a possibility. Indeed, trying to estimate the $O_2$:$CO_2$ ratio of the small net imbalance is what makes these small differences important.

Page 10 Line 17, change "values for $\alpha_B$" to "stock-based ratios"

Page 12 Lines 3-5, the sentence starting with "If our measurements. . . " should be deleted, or rewritten to state something along the lines of "In the absence of fossil-fuel influence, our measured ratios correspond to local signatures of photosynthesis and respiration. The closeness of these ratios to 1 is consistent with measurements in other temperate forests but we do not

yet have enough measurements to know if they are globally representative. Also, it is still possible that gross photosynthetic and respiratory exchanges could happen at ratios different from that corresponding to the much smaller annual net uptake of carbon and release of oxygen." You could also add "Nonetheless, our measurements provide valuable insight into the nature of $O_2$ and $CO_2$ exchange in a temperate forest and suggest that a better guess of the ratio to use in global carbon flux partitioning calculations, $\alpha_B$, may be closer to 1 than previously used. Studies that rely on $\alpha_B$ should certainly explore sensitivity to values as low as 1.0 (e.g. Resplandy et al., Science, 2018)."

Page 12 Line 17, change "are in fact globally applicable" to "match the ratio of the global net carbon sink"

Then in all these places (and maybe a few I missed): Page 7 Line 22, Page 7 Line 24, Page 9 Line 9, Page 9 Lines 12, Page 9 Line 28, Page 10 Line 13, Page 10 Line 32, Page 10 Line 2, Page 12 Line 6, Page 12 Line 10, Page 12 Line 11, Page 12 Line 12, and Page 12 Line 15, change "$\alpha_B$" to "local biotic [or photosynthetic or respiratory] flux ratios" or something similar.

2) Variability of ratios over years and seasons

These are discussed in Section 5.3 but not presented in either timeseries or seasonal cycle form. These temporal variations would help to evaluate the potential influence of fossil fuel emissions (which should have greater influence in winter), environmental drivers (is one year different from others and if so why?), and the repeatability of the ratios from day to day and year to year. I did not find the periodograms helpful in this regard and suggest replacing them with a timeseries figure of daily, weekly, or monthly ratios and also a composite seasonal cycle figure of either weekly or monthly mean ratios. It would also be helpful for at least 1 of the methods in Table 2, to show values calculated for summer and winter separately.

3) Influence of fossil fuel emissions vs respiration on day-night and low-high differences

I think that given the location of Harvard Forest, the influence of plumes of fossil-fuel pollution on the mean ratios deserves more discussion. As the authors point out in Section 5.3, the earlier study by Potasnak et al. (1999) shows fossil fuel to be a strong driver of variability in winter. Presumably this would have an effect on the all-season mean ratios presented here, but without a seasonal breakdown (see (2) above) this is difficult to assess. More specifically, I am wondering why the low ratios are systematically closer to 1 than the high ratios for both day and night and all calculation methods (Table 2). That the night-time ratios are systematically further than 1 than daytime can be explained by either respiration of high-N leaves and roots (e.g. Severinghaus, 1995 Fig. 4.1) or by greater fossil fuel influence. I think the latter is what the authors are referring to when they mention local versus regional influences in Section 5.2, and they point to fossil fuel influencing seasonal slope differences in Section 5.3, but the statements on Page 9 Lines 4-5, Page 11 Line 30 seem to discount it.

The day-night and low-high differences are really interesting, so clarifying their influences would be helpful. Specifically, one might think that the night time and lower level data would be more strongly

influenced by respiration, and that respiration might have $O_2$:$CO_2$ ratios further from 1, as speculated by Severinghous (1995). However, while the observations do show ratios further from 1 at night as might fit this pattern, they show ratios closer to 1 from the low intake, implying more fossil influences at the high level. The authors sort of state this but it could be stated more explicitly that the ratios for the night-low intake are the best estimates of a purely respiratory signature. Their differences with the day-low ratios is suggestive that the picture of Severinghaus (1995, Figure 4.1) may apply, but their range (-0.99 to -1.08) depending on method is suggestive of less N oxidation in the in situ / modern Harvard Forest soil than Severinghaus (1995) found in his older samples analyzed in lab.

Additional text (or a barplot version of Table 2) might help to highlight the observed differences. Also, it may be possible to further isolate local biotic signatures by calculating ratios of differences between the high and the low inlet at night, as remote fossil fuel influences likely affect both heights similarly (the greater fossil influence at the high intake is because of reduced biotic influence not because of differences in fossil influence). Was this tried and if so what did it show?

**Minor comments and suggestions:**

Page 1 Line 10: clarify that this number is the average and standard error of 6-hour periods.

Page 1 Line 10: Why do you expect biotic influences will dominate during the day? With photosynthesis and respiration in opposition, daytime $CO_2$ fluctuations are generally smaller than at night, when respiration is unopposed and mixing is less vigorous.

Page 1 Line 31: can you say any more in the paper about "the response of the ecosystem to environmental controls" based on interannual variations in ratios or seasonal variations (see (2) above).

Page 2 Line 10, insert "and whether N is oxidized to nitrate" after "respired"

Page 3 Line 17, if the filters are outside the traps as shown (and presumably warm) how do they trap ice crystals?

Page 3 Line 30, does it really take 4 valves to make the changeover? It would be helpful to see the individual valves in the schematic to see how this works.

Page 5 Line 12, change "lines" to "regulators" unless you have reason to specify otherwise – permeation through elastomers in the regulator is typically the reason for needing to purge. It also would be helpful to specify the regulators used.

Page 5 Line 15, it would be helpful to know when (seconds after switch) typically 70% of the change has happened. I also wonder how much the remaining 30% of rollover is affecting the signal. Did you try 80% or 90% here?

Page 5 Line 20 and Page 6 Line 26, I don't think "S2" is the official name of the scale. Please check with Ralph Keeling but I think something like the "Scripps $O_2$ Program $O_2$ Scale" is more appropriate. Also, specify WMO X2007 $CO_2$ scale (if indeed that is the scale used). Finally, missing space after "scale"

Page 5 Lines 30-31, How does a leak affect precision – variable fractionation or variable room air contamination? After tightening the packing nut did the problem go away? A leak like that could have

temperature sensitive fractionation, potentially affecting ratios – did the ratios during this period differ from before?

Page 6 Line 2 and elsewhere, I agree with Reviewer 1 that $O_2$ values should not be reported in umol/mol. Rather they should be reported in per meg, and ratios calculated after converting $O_2$ to "ppm equivalents".

Page 6 Line 6, the dominant source of variation about the fit line is likely not due to instrumental imprecision but rather real atmospheric variability from multiple sources that violate the simple model of a linear fit. Thus, a better scaling for the Deming regression would be something like 1:1 on a molar basis (or up to 1.4 to 1 if fossil fuel was the source of poor fit). This should at least be tested to see if it affects the results.

Page 6 Line 16, is the scatter from run to run not just instrumental drift over 6 hours?

Page 6 Line 31, insert "stronger influence of" before "soil"

Page 7 Lines 18-21, please clarify what is meant by "this conceptual framework generally holds." Since 1.4 is only the average fossil fuel ratio and coal at 1.17 and natural gas at 2.0 vary considerably, I would expect more than 2 end members most of the time. How variable is the $O_2$:$CO_2$ ratio for different pollution events at Harvard Forest? Also, I suggest citing Keeling dissertation (http://bluemoon.ucsd.edu/publications/ralph/34_PhDthesis.pdf) and COFFEE (https://doi.org/10.5194/acp-11-6855-201) dataset for further information of fossil fuel ratios.

Page 8 Line 3, please explain the rationale for using ½ the PBL height. Assuming fluxes mixed over the full PBL height would seem more appropriate. I don't think I've managed to fully grasp this from the Lin reference and the personal communication cited is not helpful.

Page 8 Line 14, I think this is as expected from the set up of the calculation – a better test would be to see what happens when the occasional plume of natural gas burning arrives at the site (or coal, or ocean influence). This would probably require using LPDM footprints and a spatially explicit fossil fuel emission map, or better yet a 3D transport simulation. Since both of these are likely beyond the scope here, please just acknowledge the limits to the conclusions that can be drawn here.

Section 4.2, it's not clear to me where the results of this analysis are shown. A map of average surface influence would be more helpful than a single example. Also, is the height of the particles relative to the PBL height used to calculate the region of influence? I do not think 6 particles per receptor is enough, and something like an LPDM with 100s to 1000s of particles per receptor would be more appropriate to the task. Also, since the trajectories here are only run for 6 hours I don't think their spatial range can be used to define the region of influence – locations several days back will also have influence. Without footprints defined by vertical particle locations, where to cut this off is not well defined and I don't think the statement regarding consistency with Gerbig et al. is justified.

Page 8 Line 32, what specific region is their estimate for and how did you convert this to the region around Harvard Forest – are they the same and it's just a unit conversion, or was some other downscaling needed?

Page 9 Lines 4-7, I wonder if comparing to fossil fuel influences on the basis of regional average emissions captures all of the potential impact given that fossil fuel emissions would often arrive at the

site in concentrated plumes. It might not take very many plumes of natural gas emissions at -2.0 $O_2$:$CO_2$ to affect the average ratio, if they are strong. This should be mentioned or discussed.

Page 9 Lines 11-12, some discussion and justification of the 3-sigma cutoff seems warranted. What are sigma and SEM before the cut (and perhaps add this info to Table 2)? What types of events is this cutoff meant to exclude (instrument problems, pollution plumes, low $CO_2$ variability, or all of the above) and does their exclusion sway the average ratios? You might also consider filtering based on low $CO_2$ variability to exclude ratios calculated when the denominator is small, which may be more agnostic on the actual ratios allowed past the filter.

Page 9 Line 15, I think "Further confirmation . . .  considering" should be replaced with "We also considered" as I don't think the appropriateness of this model has been confirmed and there are other ways to get ratios between 1 and 1.4 with many end members.

Page 9 Line 21 and elsewhere, somewhere please define what you mean by "steeper" and "shallower" slopes as these terms can be ambiguous.

Page 9 Lines 29-32, I think the text from "The problem with" to the end of the paragraph needs to be deleted or revised to account for the example illustrated by Severinghaus (1995), especially his Figure 4.1 in which different ratios for photosynthesis and respiration can be a permanent feature given the flux of N from leaves to soil via litter (see (3) above).

Page 10 Line 2 and elsewhere, it would also be helpful if you clarified that "larger" ratios really means more negative.

Page 10 Lines 4-8. I find the assertion that respired carbon is on average as young as weeks (here and in the Wehr and Saleska 2015 reference) somewhat puzzling. We know that about half of respiration is from soil organic matter that had to grow and die so this number must be much greater. The cited reference appears to use a general correspondence of seasonal cycles in the $^{13}C$ ratio of photosynthesis and respiration to make this claim, but this seems a bit tenuous and $^{14}C$ provides a much better estimate of carbon age. Trumbore et al. (Ecol. App. 2000, https://doi.org/10.1890/1051-0761(2000)010[0399:AOSOMA]2.0.CO;2 ) use $^{14}C$ to show that Harvard Forest soil respiration (root respiration + decomposition) has an average age of 3 years. Even allowing for a 5-25% contribution from stem respiration, this is considerably longer than the weeks mentioned here. Please try to reconcile these 2 approaches and edit the text accordingly.

Page 10 Lines 14-16, I agree with these statements but think they are indicating a detectable fossil-fuel influence on the high intake ratios and the high-low differences. It would be good to clarify this here and also acknowledge it when discussing fossil-fuel influences earlier (see (3) above).

Page 10 Line 20, please clarify how "stability might explain the shallow day-low slopes"? Is this because of less fossil-fuel influence, more photosynthesis influence, or something else? One might expect more respiration influence low and that this might lead to steeper, not shallower slopes so it would be good to spell the argument out.

**Typographical comments and suggestions:**

Many places in manuscript, including the title, there appears to be an extra space after subscripts.

I believe there should be spaces between all values and their units.

I believe units should not be italic.

Page 1 Line 9: use same number of significant digits for both subsets

Page 2, Line 1, insert "global" before "average"

Page 2 Line 6, insert "global" before "carbon"

Page 2 Lines 18-19 use "PgCyr$^{-1}$"

Page 3 Line 4, say "5 m" here but "6 m" in abstract, please clarify

Page 5 Line 7, add "seconds" after "120" (if that is what is meant), also probably don't need quotes around live since already used without.

Page 7 Line 27, add "we" before "estimate" and change "fluxes" to "contributions"

Page 8 Line 2, 22.4 corresponds to 0 C – there may be a better value or values to use here

Page 10 Line 23, "a" not "an"

Page 11 Line 12, "influences" after "anthropogenic"

Page 12 Line 3, change "molar ratio" to "all-data average molar ratio over 6-hr periods"

Page 13 Line 6, add "the" before "period"

Table 2: present the 4 metrics in the same order in each section

Figure 3: It is not possible to distinguish the 2 colors used

---

## Author Comment (AC1) · 26 Apr 2019

We thank both referees for their extremely thoughtful and valuable reviews. Our manuscript is much better as a result of addressing the questions they raised. Please see the PDF attached containing both a response to the reviews and a revised manuscript.

Please also note the supplement to this comment:
https://www.atmos-chem-phys-discuss.net/acp-2018-1041/acp-2018-1041-AC1-supplement.pdf

---

## Author Comment (AC2) · 30 Apr 2019

Response to referees:

We thank both referees for their insightful and thorough reviews. The manuscript is greatly improved as a result of their efforts.

**Referee #1:**

**General comments:**

We are well aware of the merits of the "per meg" scale for reporting O2 measurements and have used it in all of our past publications. We had chosen to present this research using mole fraction because the Local Biospheric Exchange Ratio ("$\alpha_B$" in the original draft, see our reply to comment #1 of Referee #2 below) characterizes a mole-for-mole flux balance. However, we had overlooked the importance of converting to per meg and then back to ppm equivalent. We have now done this throughout the paper and added an explanatory paragraph on units.

The second general comment suggests the use of CO and [14]C observations to assess the influence of fossil fuel combustion on our dataset. We agree completely that these would be ideal tools for quantifying the role of combustion at the forest. Unfortunately, this simply isn't an option. For most of the time we were collecting data, there were no contemporaneous measurements of either species. While CO is now routinely measured at the forest, it comes too late for our dataset. Routine [14]C measurements are prohibitively expensive. We did explore the use of $SF_6$ as an indicator of anthropogenic signals, but were convinced by Jocelyn Turnbull's work (Turnbull et al. GRL VOL. 33, L01817, 2006) that this was not a suitable tool. This is why we turned to modelling to gain insight into non-local influences.

**Specific comments:**

All suggestions have been adopted with the following exceptions and/or notes

P3, line 21-23: The tanks and regulators are now described in the caption of Table 1, as well as rough estimates of the uncertainties of the composition. Based on the intercomparison of the various tanks, we see no clear evidence of drift in the O2 composition of the tanks over the years of use. However, any quantitative estimate of stability is beyond the scope of this paper.

P6, line 7-10: Perhaps we are simply missing something important, but it seems to us that the changes in O2 and CO2 over a six-hour period will be faithfully recorded if the standard tanks are essentially constant over that same six-hour period. If the O2 drift in the tanks is (for example) 50 per meg over a year, the drift will be about 0.03 per meg over the 6-hour period from which each slope is determined. While 50 per meg drift would doom an O2-based land-sink calculation, the 6-hour drift would have no impact on the slopes. Since this paper focuses exclusively on slopes, we have chosen to omit any discussion of long term stability of the standards.

P6, line 27: Again, this kind of analysis is beyond the scope of this paper. However, your question prompted us to make a quick linear fit to the low-intake O2 time series shown in Figure 3. It does reveal a drop of about 9 ppm. Given that we have made no effort to develop a long-term calibration curve for our tanks, this is reassuring.

P7, line 29:  The Lagrangian back-trajectory model provides us with information that helps us use our 1-dim forward box model, but the two models are not linked in any specific way.  We run HYSPLIT to tell us (in general terms) how far a parcel of air might move in 6 hours, and from whence the parcels have come when they arrive at our study site.  Informed by the HYSPLIT results (in a general sense), we designed our forward model to carry a parcel of air a realistic distance over a fictitious landscape and record the influence of the idealized fluxes to/from the landscape. We have tried to clarify this with minor changes in the text.

P8, lines 3-4:  The box heights used in our 1D model are indeed derived from PBL values extracted from the NAM12 dataset.  Again, we have tried to clarify this.

P8, lines 11-12 and 16-17:  Regarding the particulars of the sensitivity tests:  Our 1D model is so simplified that we feel its output values for slopes and $O_2$ and $CO_2$ ranges are not particularly meaningful.  This is why we haven't included a table of values from the model.  What we do consider meaningful and significant is that the model is consistent with our intuition:  A mix of forest and urban fluxes usually yields a slope intermediate between 1.4 and 1.05, taller boxes yield smaller signals, and the seasonal cycle in fluxes leads to a seasonal cycle in the ranges of $O_2$ and $CO_2$ values predicted for any particular 6-hour period.  These intuitive results hold despite (largely independent) diurnal cycles in the various fluxes and the depth of the PBL.  We have changed the text to emphasize this point.

Regarding the citation of Conley (2018), the referee raises an interesting point.  While Conley (2018) is not peer reviewed, it is freely available electronically.  For this reason, we were careful not to use it as a reference for results.  Instead, we cite it only for more complete descriptions of the model, or the experiments carried out with the model.  Essentially, we have used it as an appendix or supplement to this manuscript.  If this kind of citation is inappropriate, we are certainly willing to excerpt the relevant information and turn it into a formal supplement.

Page 8, Section 4.2:  The determination of the "footprint" or "region of influence" has been a challenging part of this analysis from the beginning.  Flux footprints (which have been thoroughly studied at Harvard Forest) are quite different from the concentration footprints that are relevant to our study.  The work of Gerbig et al. (2006) is the closest match to our present needs (Figs. 2 & 4 of that paper), but even that is geared toward quantifying the cumulative influence of many small contributions integrated over a large area.   Nonetheless, Gerbig et al.  does make it clear that a 6-hour period corresponds to a primarily forested region within ~100km or less of our study site.

Page 8-9:  The calculation is indeed simple and straightforward, but it probably isn't meaningful in this case.  On account of its many simplifications, we don't trust the model enough to quantify the contributions of fossil fuels and biosphere.  As discussed above, we only consider the model a tool for testing the end-member mixing framework.   Instead, as discussed now in section 4.3, we use observationally based estimates of carbon fluxes, both biogenic and anthropogenic to argue that our summer daytime slopes are the closest thing we have to a pure biogenic signal.

Page 9, line 4-5:  We sought, and received from the authors, some clarification of ambiguous language in the Sargent et al. paper.  Based on this, and in response to questions from both reviewers, this section has been significantly revised.

Page 10, Lines 2-3:  We have substantially reworked the discussion of observations and the information gleaned from data subsets (high/low, day/night, summer/winter).  We believe this new treatment will address your question.

Page 10, Section 5.3: We have replaced the Lomb-Scargle periodograms with a year-long climatology of slopes (averaged weekly), as well as a time series of slopes for the entire study period (averaged monthly).  In addition, the discussion of observations now contains a substantial examination of the possibility of variation in the local biospheric exchange ratio (formerly $\alpha_B$).

Page 17, Figure 1:  All standard tanks were stored horizontally in an insulated blue box (for the reasons you describe).  We have changed the text and figure to indicate this.

Page 23, Figure 7:  As discussed above, we used HYSPLIT only to establish the area of influence.  This is not expected to exhibit secular variation between the beginning of our study period and the present. Furthermore, the HYSPLIT back-trajectories were not linked to our 1-D forward model runs.  Thus, any one trajectory is as good as any other and we simply chose a representative example.

Page 24, Figure 8:  Thank you for pointing this out.  The figure was left over from an earlier draft.  We have now removed it entirely.

**Referee #2 (Brit Stephens):**

1) Extrapolation to $\alpha_B$:  Essentially all suggestions incorporated.  Your comments prompted us to realize that we had been unwittingly treating $\alpha_B$ as a prognostic concept, rather than a diagnostic one.  As you correctly point out, $\alpha_B$ is a catch-all parameter, much like the "land sink" in the GCP budget is really just a collection of left-over fluxes that haven't been captured in the accounting of ocean and atmospheric measurements.  Thus, $\alpha_B$ is not something that we measure directly.  Instead, we measure something (now named the Local Biospheric Exchange Ratio or LBER in our paper) that contributes to $\alpha_B$ and with luck, is an important determinant of it. Thanks for pointing out our misrepresentation.
2) As requested, we have replaced the periodograms with time series of monthly mean values, and a separate climatological year of weekly mean values.  The text in Section 5.2 has been altered to address this with more clarity.
3) We have substantially reworked the "Discussion of Observations" section to address these concerns.  We have added figures giving graphical comparisons for average slopes calculated in various ways.  To complement this, we organized the discussion by agency (flux sizes, atmospheric dynamics, changing physiology) rather than data category (high/low, day/night, winter/summer).  We think this leads to a much more coherent examination of possible mechanisms.

Minor comments:  We have adopted your suggestions in most cases.  Exceptions, or further explanation as needed, are below:

P1 Line 10:  Changed, to avoid the use of αB and instead, refer to "local biotic influences".

P3 line 17:  Our understanding is that, in principle at least, tiny ice crystals could be entrained/suspended in the airstream and not melt/sublimate before reaching the filters (a bit like frazil ice in a river).  Perhaps this is impossible at the low flow rates in our system, but this was the thinking behind the original statement.  However, whether such ice crystals evaporate/sublimate while airborne or in the filter, they'll never raise the dewpoint much above -90C.  In any event, we have changed the text to focus on keeping particulates out of the downstream equipment.

 P3 line 30:  Because we used simple dual-action valves and wanted completely symmetric flow paths, we used four valves connected to a custom manifold.  Here is a diagram:

[Figure]

A changeover occurs when all four valves switch simultaneously from energized to de-energized (or vice versa).  We can include this diagram in the paper if such a level of detail is deemed appropriate.

P5 line 12:  We have changed the text to refer to both lines (which have stale air in them and need flushing) and regulators (which are the components that need purging, as you point out).  The details of the tanks and regulators are now given in the caption of the table listing tank compositions.

P5 line 15:  We have added the transition time, as suggested and tried to clarify our (admittedly arcane) averaging protocol.  As we now explicitly state in the text, we don't actually use all of the values that postdate the 70% transition threshold.  When developing our analysis protocol, we considered a range of post-transition "deadtimes", and finally settled on using the last ~70% of the post-transition data.  This value minimizes the standard error on each calibration run.  That is to say, working back from the end of the cal run, as we retained more and more values the standard error fell and then began to climb.  The climb was a result of gradually including remnants of the previous analyte.  Based on multiple cal runs, we find minimal standard error when using 5, 4 or 3 of the 24-second changeover-blocks of data for transition times in the first 51 secs, 103 secs or 180 secs, respectively.  If the transition happens after 180s, we discard that cal run.  This algorithm works well for all cal runs except the Low Span (LS) tank.  In

those runs, where the LS tanks has a $CO_2$ mixing ratio roughly 100ppm higher than the working tank (WT), 6 minutes is simply insufficient for the system to come to equilibrium. Consequently, all LS-WT comparisons are biased low. For our 6-hour slope calculations, this bias is not a problem. Looking ahead to work on low-frequency variations and intercomparisons with other labs, we are developing ways to correct for this bias.

P5 line 20 and P6 line 26: The $CO_2$ scale is now correctly referenced, as suggested. The $O_2$ scale is referred to as the "S2" scale by Keeling et al. 2007. When originally preparing this manuscript we did indeed check with Ralph Keeling and this was his preferred name for the scale (and preferred citation).

P5 lines 30-31: You have (perhaps inadvertently) drawn attention to a mistake in our reasoning. The statement about the loose packing nut was true, but we realize now it was likely irrelevant. The crossover valve is only involved in the intake of sample air. However, the instrumental performance shown in Figure 2 is based on standard tanks. Thus, while a loose packing nut might well degrade analyses of samples, it can't explain noisy standard data. At this point, our best explanation for poor standard tank precision is that a leak developed around or within the 6-port selector (Valco) valves. We won't be able to test this hypothesis until the instrument is fully operational again. Whether the tightening of the packing nut changes sample precision also remains to be seen, as the instrument has not been operating since we made that repair. In the meantime, we have adjusted the text accordingly, and omitted mention of the loose packing nut.

P6 Line 2: Done. Please see response to reviewer #1.

P6 Line 6: In this case, we believe the uncertainties on the measurements are the right quantities to use in the Deming regressions. It is certainly true that atmospheric variability from one measurement to the next is much larger than the uncertainties, but unlike analytic errors, the atmospheric variability in $O_2$ and $CO_2$ is correlated. Whatever the reason ambient $O_2$ might be high or low from one moment to the next, $CO_2$ will be low or high. This is the essence of the linear model we are applying (with the slope as a free parameter) and fundamentally, there's no difference between these variations and phenomena we are trying to measure. In contrast, the errors are uncorrelated, non-negligible and unequal for the two species.

P6 Line 16: We have substantially rewritten this subsection and changed our approach to calculating the errors on our measurements. Before, we had started with the scatter on a single calibration tank (and then compared this with inter-tank scatter). Now, we start with inter-tank scatter and work backward to errors on single measurements, which we then use to estimate errors on the atmospheric values. We believe this is a more conservative approach, since inter-tank scatter captures additional potential sources of systematic error.

P8 Line 3: We have changed the text to explain how ½ the PBL height is used in the STILT model. Since we use STILT to determine the provenance of our signal, we use ½ the PBL height in our own 1-D model for consistency. Furthermore, Eq. 4 shows that changing the box height would change the magnitude of all of our signals in the same way. Thus, while a better value for box height might give different slopes, and would certainly give different ranges of variability for each species, it would make no difference to the conclusion we draw from the 1-D model: A simple mix of fluxes gives a simple mix of slopes.

Section 4.2:  Our guess is that we are using "region of influence" differently than you are.  We aren't trying to identify all parts of the country that contribute to the observed concentrations at Harvard Forest.  This would indeed extend over an enormous area.  Instead, we are asking where is the parcel of air at t = 0 that will be at our study site at t = 6hr (the end or a sampling interval).  Likewise, where is the parcel of air at t=-6 that will be at our site at t=0.  For this reason, we've replaced "region of influence" with "6-hour provenance".  Naturally, the composition of the air at t=0 was set by the processes upwind dating back to t= -6hr.  We try to capture this with model spin-up.  We also assume that during the 6-hour interval (t = -6 through t=0), the composition of air arriving in the box farthest upwind is essentially constant.  This assumption is supported by the relatively constant wind direction and speed we see in the HYSPLIT back-trajectories (Fig. 7).   Furthermore, if we had big shifts in wind direction bringing big changes in the "input" concentration, we wouldn't have the nice linear relationships we see evolving in most of our O2 vs. CO2 plots.  Please see also our response to Referee #1, above.

P8, line 32:  We sought, and received from the authors, some clarification of ambiguous language in the Sargent et al. paper.  Based on this, and in response to questions from both reviewers, this section has been significantly revised.

P9, lines 4-7:  As we mention above in reply to your question on P8, line 32, we have revised this section of the text with more specific information.  You are correct that a concentrated plume of combusted natural gas would skew our results.  Referring to ACES and the EPAs Greenhouse Gas Reporting Program (https://ghgdata.epa.gov/ghgp/main.do)  there are few power plants in the general area of Harvard Forest, and fewer still that are natural gas fired.  Nonetheless, without contemporaneous CO or C2H2 data, we can't rule out this possibility so we have made a brief mention of it.

P9 Lines 11-12:  The choice to employ a 3σ iterative cut was the outcome of much study and deliberation.  To include documentation of all the supporting work (even in an appendix) would be excessive.

Fundamentally, we are just trying to get rid of "bad slopes": slopes that are poorly constrained by the data or that are unduly influenced by a small subset of anomalous points.  Bad slopes are pretty easy to identify visually, but with ~3700 6-hour high- and low-intake subsets, this is simply out of the question.  When we actually look at some of the slopes with unphysical values (>0 or <-2), they have extreme values for many reasons:  one point with very low O2 but reasonable CO2 (presumably an Oxzilla glitch), low variability in both O2 and CO2 (yielding a cluster of points with no clear correlation), etc.  We tried to flag these slopes using statistical indicators of fit quality ($\chi^2$ and statistical uncertainty in the slope) as well as scatter plots of slope vs. CO2 range and slope vs. O2 range but found nothing particularly useful.  You are correct that periods with low CO2 variability tend to have a wider range of slopes, but plenty of these data subsets actually have tight O2-CO2 correlations.  Furthermore, a cut on CO2 range would not be entirely agnostic, as there does seem to be a weak correlation between CO2 range and slope, and this correlation changes with day and night, summer and winter.

For these reasons, we adopted the iterative cut.  As Figure 6 shows, there are significant non-gaussian tails even after it is employed so it's clear the cut is not too draconian.   While a discussion like the one above is inappropriate for the paper, we have added prose about the purpose of the cut and we have added values to Table 2 that clarify the impact of the cuts.

Page 9, lines 29-32:  We agree that the flux of nitrogen from leaves to soil will have an influence on LBER.  The question is one of timing and duration of disequilibrium.  Even in the idealized scenario outlined in section 4.1.2, Severinghaus explicitly acknowledges that the system must eventually yield the same average exchange ratios for photosynthesis and respiration due to mass balance.  Instead of his Fig. 4.1, imagine a pair of trees.  One of them (shown in Fig. 4.1) is standing, and PQ for the tree is lower than OR for the soil.  The other tree has fallen over and its trunk is decaying, with OR lower than PQ.  In other words, to invoke different exchange ratios for day and night requires an ecosystem that (in this simplified example) only has standing trees.  Given that Harvard Forest is measured to be a net carbon sink (i.e. a system with photosynthesis and respiration unequivocally out of balance) it is possible that Severinghaus's original Fig. 4.1 is a better analogue for Harvard Forest than the two-tree picture we suggested above, but it is also possible that this patch of forest is diverse enough in age and organismal composition, that it contains both standing and fallen trees (so to speak).  We have changed the text to acknowledge these two possibilities.

In a more realistic analogy, there isn't one standing tree; there are several, each with a different age and a different elemental ratio.  These represent the many pools into which carbon may be allocated (leaves, roots, stem wood, etc.) and their various carbon residence times.  Again, to invoke different day/night exchange ratios requires the "standing trees" to be different from the "fallen trees".

Page 10, lines 4-8: This is an interesting question and led us to correspond with Rick Wehr.  He pointed out that more recent data (Wehr et al 2016) show the cycle of respired 13C is somewhat damped, relative to the photosynthetic cycle.  This suggests that the Wehr and Trumbore work might be consistent if most respired material was synthesized only days before (thus, no visible lag in the seasonal cycle), but a substantial amount of respired carbon came from a much older pool with no seasonality in its 13C signature (causing damping of the respired 13C cycle).  Wehr points out that Trumbore's values have no uncertainties, making it hard quantify (in)consistency.

This analysis is reinforced by an unusual 14C-based study described by Trumbore et al. (Eos V.83 N.24, 2002) in which they say "Previous studies with deliberate 14C labeling *[Hanson et al.,* 2000] have shown that >90% of the added 14C may be respired within a few days, with <10% allocated to longer-lived carbon pools such as leaves, roots, and wood".

We have added a brief discussion of these issues to the text.

**Typographical comments and suggestions:  All done except…**

Page 1, Line 9:  The two datasets in question are very different sizes.  We have significantly fewer data in the "summer, low, day" subset, so precision is reduced.  For this reason, we prefer to leave the presentation unchanged.

Page 8, line 2:  We have stuck with 22.4 but acknowledged the potential problems with this value.

[revised manuscript text omitted]